Review Article

EMBO
Molecular Medicine

# New frontiers in prostate cancer treatment from systemic therapy to targeted therapy

Shaghayegh Nouruzi[1,2], Maxim Kobelev[1,2], Nakisa Tabrizian[1,2], Martin Gleave [1,2] & Amina Zoubeidi [1,2✉]

## Abstract

**Significant advances in prostate cancer (PCa) treatment have occurred through the integration of molecular biomarkers and imaging with targeted therapies. While androgen receptor pathway inhibition (ARPI) remains the cornerstone of PCa therapy, the current therapeutic landscape has expanded to include a broader range of targeted agents, alongside emerging approaches that leverage disease-specific vulnerabilities. Molecular profiling has enabled the exploration of diverse therapeutic modalities, including epigenetic regulators, immune-modulating agents, metabolic pathways, kinases, and cell surface proteins. Despite this progress, further research is needed to address tumour heterogeneity and treatment-resistant phenotypes. As ARPI use moves earlier in the disease course and novel agents are incorporated into standard care, prolonging disease control may also reshape emergent resistant phenotypes and disease progression trajectories. This evolving context underscores the need to revisit agents that may now show efficacy in new therapeutic settings or when paired with complementary strategies. Here, we review the current treatment framework in PCa and highlight novel approaches and targets poised to transform clinical care.**

**Keywords** Prostate Cancer; Targeted and Systemic Therapies
**Subject Categories** Cancer; Urogenital System

## Introduction

The treatment landscape for prostate cancer (PCa) has advanced significantly, providing effective strategies for both localized and metastatic disease. As a hormone-driven malignancy, PCa relies on androgens and androgen receptors (AR) for growth and survival. For localized disease, surgery and radiation therapy remain the standard of care. In advanced, recurrent, and metastatic cases, androgen deprivation therapy (ADT), AR pathway inhibitors (ARPIs), and chemotherapy play dominant roles in PCa treatment. Despite their initial efficacy, most patients eventually progress to the castration-resistance stage, often reactivating the AR pathway

(Kirby et al, 2011) or shifting to alternative lineages independent of AR (Kregel et al, 2016).

In AR-driven castration-resistance PCa (CRPC), key resistance mechanisms include AR mutations that enable activation by alternative steroid hormones, AR amplification that increases receptor sensitivity (Fujita and Nonomura, 2019; Gottlieb et al, 2012; Linja et al, 2001; Robinson et al, 2015a), tandem duplication of AR enhancers (Quigley et al, 2018; Viswanathan et al, 2018), splice variants like AR-V7 (Cao et al, 2014) that allow AR to remain active without a ligand-binding domain (Guo et al, 2009; Li et al, 2013), androgen biosynthesis (Chang et al, 2013; Green et al, 2012) and alteration of AR co-factors. Beyond AR, alterations in clinically actionable targets in the DNA damage repair pathway, phosphoinositide 3-kinase (PI3K) signaling, and high microsatellite instability (Abida et al, 2019a, Robinson et al, 2015b) provide additional therapeutic avenues for patients.

AR-independent subtypes like neuroendocrine PCa (NEPC), double-negative PCa (DNPC) and amphicrine (Aggarwal et al, 2018; Beltran et al, 2019; Beltran et al, 2016; Davies, Zoubeidi et al, 2023; Kim et al, 2024) had emerged through lineage plasticity. These variants are enriched with TP53, RB1, and PTEN loss and characterized by poor prognosis, typically treated with platinum-based regimens, taxanes, and etoposide (Akamatsu et al, 2018; Alabi et al, 2022; Beltran and Demichelis, 2021; Vlachostergios et al, 2017) with survival often under two years. These subtypes are often driven by their unique epigenetic and transcriptional control (Davies et al, 2023; Sychev et al, 2024), which may represent vulnerability.

Building on the advances over the past two decades, the future of PCa treatment lies in adopting more personalized approaches, with therapies designed to address each patient's unique molecular, genetic, epigenetic, and transcriptomic profiles. As we unravel the complexities of resistance mechanisms and tumor heterogeneity, developing targeted therapies and strategic combinations will be critical to overcoming resistance, controlling metastatic progression, and improving long-term survival. Here, we will explore the spectrum of PCa treatments, from the current standard of care and established systemic therapies to emerging innovations, evaluating successes and failures along the way. New therapeutic targets and approaches are continuing to emerge, ranging from AR-targeted therapies, radiotherapy, immunotherapy, epigenomic and metabolic modulation, inhibition of kinases, and cell surface-directed strategies (Fig. 1). By understanding where we stand and where we are headed, we aim to shed light on the evolving landscape of PCa care.

[1]Department of Urologic Sciences, University of British Columbia, Vancouver, BC V5Z 1M9, Canada. [2]Vancouver Prostate Centre, Vancouver, BC V6H 3Z6, Canada.
✉E-mail: azoubeidi@prostatecentre.com

**Glossary**

| Term | Definition |
|---|---|
| ADC (Antibody–Drug Conjugate) | A targeted therapy combining an antibody with a cytotoxic drug to deliver treatment directly to cancer cells. |
| ADT (Androgen Deprivation Therapy) | A treatment that reduces androgen levels or blocks androgen receptor (AR) activity. |
| AR (Androgen Receptor) | A nuclear receptor that regulates gene expression in response to androgens and plays a central role in prostate cancer. |
| ARPI (Androgen Receptor Pathway Inhibitor/Inhibition) | Refers to either drugs that inhibit the androgen receptor signaling pathway (e.g., enzalutamide, abiraterone) or the therapeutic strategy of suppressing AR activity. The term is used contextually to describe both the inhibitors themselves and the broader concept of pathway inhibition. |
| BAT (Bipolar Androgen Therapy) | A therapeutic approach involving cycles of androgen deprivation followed by supraphysiologic androgen administration to modulate AR signaling. |
| BCR (Biochemical Recurrence) | A rise in prostate-specific antigen (PSA) levels after initial treatment, indicating potential cancer recurrence. |
| BiTE (Bispecific T Cell Engager) | A type of immunotherapy that links a tumor antigen to CD3 on T cells, bringing them into close proximity to cancer cells to trigger targeted T cell–mediated killing. |
| BRCAness | A phenotype where tumors exhibit defects in HRR despite not having BRCA1/2 mutations, potentially making them sensitive to PARP inhibitors. |
| CAR T (Chimeric Antigen Receptor T Cell) | A type of cell therapy where a patient's T cells are genetically engineered to express a synthetic receptor that targets a specific tumor antigen, enabling them to recognize and kill cancer cells. |
| CRPC (Castration-Resistant Prostate Cancer) | Prostate cancer that continues to progress despite androgen deprivation therapy. |
| CSPC (Castration-Sensitive Prostate Cancer) | Prostate cancer that still responds to treatments that lower androgen levels. |
| ctDNA (Circulating Tumor DNA) | Fragments of tumor-derived DNA found in the bloodstream, used for non-invasive cancer diagnostics. |
| DNA Methylation | An epigenetic modification involving the addition of a methyl group to cytosine bases, typically at CpG sites. |
| DNPC (Double-Negative Prostate Cancer) | A subtype of prostate cancer lacking both AR signaling and neuroendocrine features. |
| Histone Modifications | Reversible changes (such as acetylation, methylation, phosphorylation) to histone tails. These modifications influence chromatin structure and regulate gene expression. |
| HRR (Homologous Recombination Repair) | A DNA repair pathway that fixes double-strand breaks; often disrupted in cancer. |
| ICB (Immune Checkpoint Blockade) | Immunotherapy targeting checkpoints like PD-1/PD-L1 to restore anti-tumor immune responses. |
| MMR (Mismatch Repair) | A DNA repair system that corrects errors introduced during DNA replication. Deficiencies in MMR can lead to microsatellite instability and increased mutation rates. |
| MSI (Microsatellite Instability) | A condition of genetic hypermutability resulting from impaired mismatch repair (MMR). MSI-high tumors often have better responses to immune checkpoint blockade therapies. |
| NEPC (Neuroendocrine Prostate Cancer) | An aggressive, AR-independent subtype of prostate cancer characterized by neuroendocrine differentiation. |
| OS (Overall Survival) | The length of time from diagnosis or treatment start until death from any cause. |
| PCa (Prostate Cancer) | A malignancy originating in the prostate gland, often driven by androgen signaling. |
| PRAD (Prostate Adenocarcinoma) | The most common histological subtype of prostate cancer, typically characterized by a luminal epithelial and AR-driven phenotype. |
| PROTAC (Proteolysis Targeting Chimera) | Molecules that induce targeted protein degradation via the ubiquitin–proteasome system. |
| rPFS (Radiographic Progression-Free Survival) | The time during which a patient lives without radiographic evidence of disease progression, typically measured by MRI or PSMA-PET imaging. |
| RLT (Radioligand Therapy) | A therapy that uses radioactive ligands to target and destroy cancer cells expressing specific surface markers. |
| TriTACs (Trispecific T-cell Activating Constructs) | A next-generation class of T-cell engager molecules designed to target cancer cells by binding to a tumor antigen, CD3 on T cells, and albumin to extend half-life. TriTACs enhance T cell–mediated cytotoxicity and are being developed as a more durable and selective alternative to BiTEs. |
| Tumor Microenvironment (TME) | The complex and dynamic environment surrounding a tumor, including immune cells, stromal cells, blood vessels, signaling molecules, and extracellular matrix. |

# Advances in targeting AR pathways

ADT reduces circulating androgen levels or blocks AR activity to inhibit the growth of androgen-dependent PCa. This can be achieved through surgical castration (bilateral orchiectomy), medical castration using gonadotropin-releasing hormone (GnRH) analogs or antagonists, or the use of AR pathway inhibitors (Crawford et al, 2019; Mansinho et al, 2018; Mohler et al, 2016). ADT is typically the first-line systemic treatment for PCa (Heidenreich et al, 2014; Mohler et al, 2016).

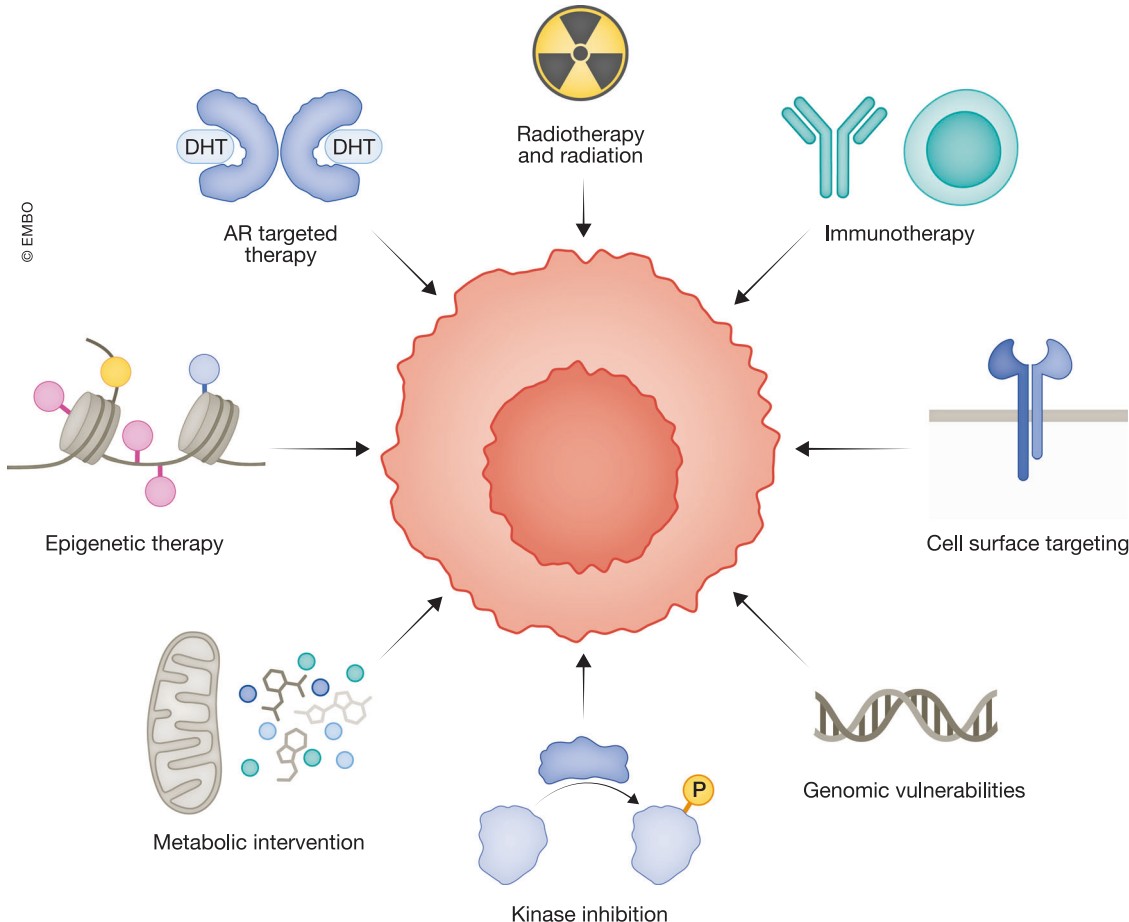

**Figure 1. Therapeutic strategies targeting prostate cancer (PCa) vulnerabilities.**

A schematic representation of diverse treatment approaches in PCa. Current and upcoming key therapeutic strategies include androgen receptor (AR)-targeted therapy, radiotherapy and radiation, precision therapy exploiting DNA repair deficiencies and other actionable mutations, immune checkpoint blockade and adoptive T cell therapies, targeting chromatin modifiers and epigenetic regulators, antibody–drug conjugates, BiTEs, and radioligands directed against cell surface proteins, blocking oncogenic signaling cascades involving kinases, and disruption of metabolic pathways.

AR pathway inhibitors targeting the ligand-binding domain (enzalutamide (ENZ), apalutamide (APA), darolutamide (DARO)) and the de novo synthesis of androgens (abiraterone acetate (ABI)) have changed the treatment landscape across several PCa disease states. ARPI is the current standard of care in 1st line metastatic castration-resistant PCa (mCRPC), in combination with ADT in metastatic castrate-sensitive PCa (CSPC) (Armstrong et al, 2022; Chi et al, 2019; Sternberg et al, 2020) and high-risk biochemical recurrence (BCR) post-surgery or radiotherapy (Achard and Tombal, 2024). For instance, the EMBARK (ADT + ENZ) and PRESTO (ADT + APA) trials provided proof of principle that treatment intensification is superior to ADT in non-metastatic (M0) CSPC and high-risk BCR (2023; Freedland et al, 2021). A similar trial combining ADT and DARO (ARANOTE) is ongoing in mCSPC and will likely show similar results. Furthermore, the PROSPER trial showed that the addition of ENZ to ADT significantly improved overall survival (OS) in M0 CRPC (Sternberg et al, 2020).

Novel agents targeting de novo synthesis of androgens are currently under clinical investigation. PRL-02, an injectable prodrug of abiraterone, is in a phase I clinical trial in mCSPC and mCRPC (NCT04729114). MK-5684, an inhibitor of CYP11A1 (catalyzes the first step of steroid-hormone biosynthesis) (Karimaa et al, 2022), has shown evidence of anti-tumor activity in heavily pretreated mCRPC (Fizazi et al, 2024). Two phase 3 clinical trials OMAHA1 (NCT06136624) and OMAHA2 (NCT06136650), currently evaluate MK5684 in heavily pretreated mCRPC patients or those who have previously failed one ARPI, respectively, evaluating OS and radiographic progression-free survival (rPFS) of MK5684 compared to ENZ or ABI. If approved, this drug may give patients another treatment option after progression with standard of care ARPI. Moreover, a previous landmark study has demonstrated that intermittent ADT allows testosterone recovery and improves tolerability without compromising outcomes compared to continuous treatment (Crook et al, 2012). The ongoing LIBERTAS trial will inform whether intermittent ADT plus APA in mCSPC can improve the side effects of ADT without compromising rPFS compared to continuous ADT plus APA (NCT05884398).

Other modalities to target AR are under development. For instance, targeting the AR N-terminal domain (NTD) offered an

alternative strategy. EPI-7386 showed acceptable safety profiles in phase I trials (NCT04421222, NCT05075577), but phase II trials in combination with ENZ in mCRPC patients were terminated due to a lack of improved efficacy over ENZ alone. This drug is also being evaluated in combination with ENZ in mCSPC patients (NCT06312670). Nevertheless, efforts to develop next-generation NTD-targeting agents are ongoing and may yet provide new avenues for overcoming resistance. The next generation of AR-targeted therapies is AR degraders, which should exhibit superior AR pathway inhibition compared to AR antagonists, combat AR-mediated resistance mechanisms, and offer a third line of ARPI following progression on ENZ/ABI. Currently, AR PROTACs are under evaluation in phase I and III in mCRPC (NCT05177042, NCT06764485). However, it is still unclear whether such potent ARPI will manifest in higher rates of AR-independent tumors.

A counterintuitive approach to battling PCa is androgen bipolar therapy (BAT), which cycles testosterone levels between supraphysiological and castrate levels. Preclinical studies have demonstrated that BAT may sensitize tumors to subsequent ARPIs (Sena et al, 2021), inhibit proliferation (Vander Griend et al, 2007), down-regulate MYC expression (Sena et al, 2022), and induce DNA damage (Chatterjee et al, 2019). In addition, it activates the immune signaling pathways, potentially enhancing the response to checkpoint inhibitors (Markowski et al, 2024). Clinical trials have demonstrated improved quality of life with BAT compared to standard care (Markowski et al, 2024). The TRANSFORMER trial comparing BAT and ENZ reported comparable rPFS and OS (Sena et al, 2021). Ongoing trials are investigating BAT combined with chemotherapy, Radium-233, and ENZ in mCRPC patients (NCT06039371, NCT04704505, NCT04363164). Many questions remain before BAT becomes the standard of care:

a. Which patients benefit from the addition of BAT compared to continuous standard of care?
b. Does BAT alter the rate of treatment resistance or the landscape of treatment resistance mechanisms?
c. Which drug combinations can synergize with the anti-tumor mechanisms of BAT?

Numerous agents are targeting the AR pathway, but how can we explore novel combinations and restructure approaches within our existing toolkit to enhance outcomes and improve quality of life? As ARPIs are brought earlier in the treatment landscape, we should be wary of speeding up the emergence of treatment resistance mechanisms and running out of treatment options. The trials exploring the use of novel agents to tackle ARPI resistance show promising results, but historically, we learned that resistance to targeting AR will ultimately occur. The challenge will be, what are the available drugs that benefit these patients if these new agents fail? Will AR-independent subtypes dominate the landscape, or will we give rise to new phenotype(s)? A key tool that will help to overcome this challenge will be routine use of circulating tumor DNA (ctDNA) sequencing to track the emergence of AR-independent lineage and adjust treatments accordingly (De Sarkar et al, 2023).

## Radiotherapy and radiation therapy

Radiotherapy is a mainstay treatment modality for localized PCa. For low-risk patients, brachytherapy is a common choice (Peinemann et al, 2011), while hypofractionated RT and stereotactic body radiotherapy (SBRT) offer shorter effective treatment durations. These innovations are expanding the range of treatment options, making radiotherapy a versatile approach for managing PCa (van As et al, 2024; Widmark et al, 2019). A recent clinical trial showed SBRT was well-tolerated after radical prostatectomy compared to conventionally fractionated radiotherapy (Nikitas et al, 2025). Radiotherapy can also be used in certain advanced PCa patients. Radium-223 ($^{223}$Rd) was the first treatment approved to extend OS in mCRPC by targeting bone metastases without visceral involvement (Morris et al, 2019). $^{223}$Rd emits high-energy alpha particles that induce DNA damage in the tumor while sparing normal tissues. It is effective as a monotherapy (Parker et al, 2013) or in combination with ARPIs (Parker et al, 2013; Sartor et al, 2018). Although $^{223}$Rd demonstrated efficacy alone, a recent study showed that $^{223}$Rd has superior efficacy when combined with ENZ or ABI (Petrylak et al, 2021). Ongoing phase III trials are now comparing the combination of $^{223}$Rd and ENZ to ENZ alone (NCT02194842), and $^{223}$Rd and docetaxel (DOC) to DOC alone (NCT03574571) to evaluate if the addition of $^{223}$Rd improves rPFS for mCRPC patients. Of note, emerging strategies include combining $^{223}$Rd with DNA-PKc inhibitor to overcome DNA damage resistance or PD-L1 inhibitor to promote immune-mediated killing in advanced PCa patients who progressed on ARPI or chemotherapy are ongoing (NCT04071236).

## Actionable genomic alterations

The multifocal and heterogeneous nature of PCa highlights the importance of comprehensive genomic profiling to identify targetable alterations, refine treatment strategies and predict treatment response. Serial profiling across disease stages reveals an evolving mutational landscape with significant mutational diversity within each state, which may contribute to the varying clinical progression of the disease (Barbieri et al, 2013; Ku et al, 2019; Marino et al, 2023). Identification of these targets is limited by insufficient sampling, tissue access, and timing with respect to treatment.

### Homologous recombination repair (HRR)

Mutations in HRR genes are increasingly recognized to be associated with the development, aggressiveness, and ARPI sensitivity of PCa. When HR repair is disrupted, tumors become heavily reliant on single-stranded break (SSB) repair through poly-adenosine diphosphate-ribose polymerase (PARP), making them particularly vulnerable to PARP inhibitors (Fig. 2). Olaparib and rucaparib, both approved PARP inhibitors, have shown significant survival benefits in patients with HRR-deficient (HRRd) mCRPC (Abida et al, 2023; Golan et al, 2019; Mateo et al, 2015; Moore et al, 2018). Other PARP inhibitors have also been evaluated in phase II clinical trials for mCRPC. For example, the TALAPRO-1 and GALAHAD trials evaluated PARPi in patients who progressed on taxane-based chemotherapy and ARPIs and showed anti-tumor activity (Mehra et al, 2022; Smith et al, 2022). Early results showed that the combination of PARPi with ARPIs as first-line therapy has substantial benefit for mCRPC with BRCA1/2 or ATM HRR (Hussain et al, 2024).

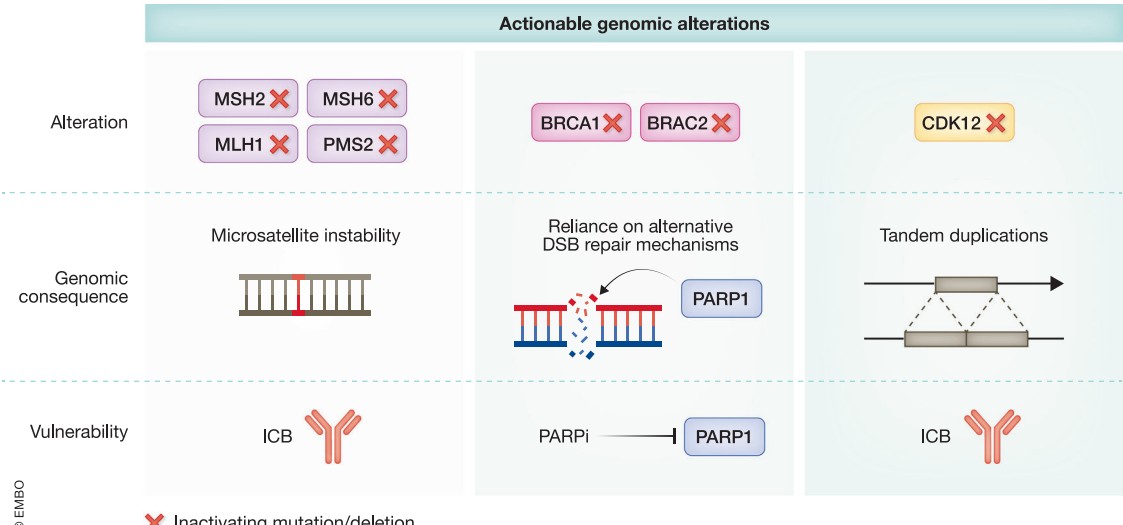

**Figure 2. Clinically actionable genomic alterations in PCa.**

Key genomic alterations inform precision treatment approaches. Mismatch repair (MMR) deficiency, involving genes such as *MSH2, MSH6, MLH1*, and *PMS2*, leads to microsatellite instability (MSI) and a hypermutated phenotype, which may be targeted by immune checkpoint inhibitors (ICB). Homologous recombination repair (HRR), often due to *BRCA1* or *BRCA2* mutations, sensitizes tumors to PARP inhibitors (PARPi). CDK12 loss, characterized by tandem duplications, is another biomarker of genomic instability associated with potential ICB sensitivity.

Interestingly, several clinical trials aim to assess moving PARPi earlier in the treatment landscape. The AMPLITUDE trial is assessing niraparib in combination with ABI in mCSPC vs ABI alone (NCT04497844). The TALOPRO trial demonstrated that talazoparib plus ENZ improved patient outcomes vs ENZ alone in mCRPC patients, regardless of HRR gene alteration status, although the benefit was greater in HRRd patients (Agarwal et al, 2023), and is now evaluating its combination with ABI in mCSPC (NCT04734730). Ongoing phase I/II in mCSPC/mCRPC (NCT05367440) and phase III in mCSPC (NCT06120491) trials are exploring a novel PARPi in combination with ARPIs. Next-generation of PARPi, such as AZD5305, is now in phase III clinical trial in mCSPC in combination with ARPIs (NCT06120491). Notably, the TRIUMPH trial established that PARPi without ADT in mCSPC had no clinical benefit (Markowski et al, 2023), emphasizing the importance of combination with ARPIs. In vitro experiments have shown that AR inhibition induces an HRRd-like phenotype, offering a potential mechanistic link to why concurrent ARPI is required for PARPi efficacy (Agarwal et al, 2023).

HRR-proficient mCRPC patients derive less benefit from PARPi, raising concerns about biomarker sensitivity. False negatives may occur due to limitations in tissue sampling, tumor heterogeneity, or technical issues in detecting HRRd. Therefore, beyond identifying HRRd, there is a growing need for more functional assays that assess defective HRR activity to better stratify patients and predict response. Furthermore, other alterations may induce a state of BRCAness, which could lead to PARPi sensitivity (Xavier et al, 2021), which may also explain why some HRR-proficient patients still derive benefit. A more comprehensive suite of biomarkers will broaden the pool of patients who may benefit from PARPi, While PARPi has initial efficacy in HRRd (in particular BRCA alterations) PCa, resistance develops as tumors can adapt to treatment through mechanisms including PARP mutation, BRCA reversion mutations,

or loss of other DNA damage response pathways (Dias et al, 2021; Li et al, 2020; Seed et al, 2024; Teyssonneau et al, 2021). Emerging strategies to counter resistance are underway. Preclinical studies suggest combining PARP and ATR inhibitors (Tsujino et al, 2023), while clinical trials are investigating the combination of PARPi with inhibitors of another DNA repair mechanism (Polθ) to induce synthetic lethality (NCT06560632). Although biomarker-driven approaches like cfDNA helped tailor treatment and select patients who benefited from PARPi (Carreira et al, 2021; Goodall et al, 2017), tracking the treatment response through cell-free DNA may help identify resistance earlier.

## Mismatch repair (MMR)

MMR deficiency (MMRd) is present in roughly 5% of high-grade PCa, (Abida et al, 2019a) and results in high tumor mutation burden (TMB) and microsatellite instability (MSI). These biomarkers are now powerful predictors of clinical benefits from immune checkpoint blockade (ICB) in advanced PCa (Fig. 2). Although previous ICB trials in PCa have been unsuccessful, retrospective analysis has shown a clinical benefit for MSI-high patients. A clinical trial of anti-PD1 compared to standard-of-care chemotherapy in MSI-high metastatic colorectal cancer showed improved patient outcomes with ICB (Graf et al, 2022). Given the evidence in multiple cancers (Andre et al, 2020; Oliveira et al, 2019) and PCa, the FDA has approved the use of pembrolizumab for tumor-agnostic use in MSI-high patients, including PCa.

## CDK12 alterations

CDK12 is a cyclin-dependent kinase involved in transcriptional regulation of DNA damage response genes, particularly those required for HRR, such as BRCA1 (Blazek et al, 2011; Wu, Yu et al,

2023). CDK12 loss-of-function mutation occurs in 5–10% of advanced PCa cases and leads to genomic instability and tandem duplicator phenotype (Quigley et al, 2018; Wu et al, 2018). These alterations create neoantigens that could enhance response to ICB (Wu et al, 2018) (Fig. 2). Despite the link of CDK12 mutation to HR deficiency (Chou et al, 2024) and genomic instability, these patients showed minimal activity to ICB (Antonarakis et al, 2020; Nguyen et al, 2024) or PARPi (Antonarakis et al, 2020). Continuous efforts to investigate pembrolizumab (PD1 inhibitor) in mCRPC with either mutation in MMR or biallelic inactivation of CDK12 are ongoing (NCT04104893). Moreover, preclinical studies suggest that combining CDK12/13 inhibitors (Tien et al, 2024) with AKT inhibitors could induce synthetic lethality (Chang et al, 2024).

Understanding how alterations in the cancer genome influence treatment response can facilitate more personalized approaches. Routine use of genomic biomarkers is key to bridging gaps in understanding how genomic alterations influence disease biology and improve patient stratification to enhance outcomes. For instance, the top alterations in advanced PCa are AR, TMPRSS-ERG fusions, TP53, PTEN, RB1, and FOXA1, yet our understanding of dependencies associated with these genomic alterations remains limited, and only a small fraction of these alterations are actionable. Beyond its role as a tumor suppressor, TP53 regulates chromatin organization and transcription (Serra et al, 2024). Similarly, RB1 safeguard cell cycle progression, but also modulates E2F1 activity (Mandigo et al, 2021), as well as AR signaling in CRPC (Graf et al, 2022). Understanding these downstream effects can uncover new vulnerabilities and guide the development of more effective, targeted interventions.

A powerful and minimally invasive approach is ctDNA profiling, which may identify mutations in key genes such as TP53, RB1, SPOP, PTEN, and AR to inform prognosis and androgen dependencies (Annala et al, 2018; Swami et al, 2022a). It also showed a pivotal role in identifying truncating BRCA2 defects, enabling the prediction of vulnerabilities to PARP inhibition (Chi et al, 2022; Clarke et al, 2022; de Bono et al, 2020; Fizazi et al, 2023). While ctDNA offers a powerful, non-invasive approach to detect and characterize patient tumors, there are still limitations. Some patients with low or undetectable ctDNA can still have cancer cells present that will progress to mCRPC. It is challenging to determine alterations in the tumor genome with low ctDNA fractions (<2%) (Fonseca et al, 2024). Alteration detection is further limited by the sequencing methodology used, such as targeted panels, whole-exome, and whole-genome sequencing. ctDNA may also miss some alterations that would otherwise be detected by sequencing the tumor tissue directly (Iams et al, 2024; Vandekerkhove et al, 2021). Ultimately, multi-modal approaches will be required to characterize tumors and guide optimal treatment options. ctDNA should be complemented with transcriptomic, IHC, or radiographical imaging, including PET (Fiala et al, 2022; Lennon et al, 2020).

# Re-evaluating immune-based therapies in prostate cancer

Sipuleucel-T approval for treating metastatic PCa (Cheever and Higano, 2011) marked the start of systemic immunotherapy as a treatment option for CRPC. After more than a decade, efforts to harness immune-based therapies have faced significant challenges,

with no breakthroughs reaching the clinic. The impact of ICB in PCa has been limited (Ribas and Wolchok, 2018), largely due to an immunosuppressive tumor microenvironment (TME), low mutational burden, and minimal immune cell infiltration (Bou-Dargham et al, 2020). A major barrier to ICB and Chimeric Antigen Receptor (CAR) T-cell therapy is a T-cell-depleted TME (Bian et al, 2024) and enrichment of anti-inflammatory M2 macrophages, contributing to immune suppression (Zarif et al, 2019). Despite these obstacles, research is ongoing to elucidate the complex immune landscape of PCa, and identify predictive biomarkers of immunotherapy response, as well as combinations with approved and novel drugs that enhance the efficacy of immunotherapy.

## Immune checkpoint blockade

ICB works by removing the inhibitory mechanisms on T-cells and enhancing their cancer-fighting capabilities. However, due in part to low T-cell infiltration, ICBs have shown poor efficacy in PCa (Ozbek et al, 2022; Stultz and Fong, 2021). To improve outcomes, ongoing trials are testing ICBs in combination with standard of care treatments, including ARPIs, taxanes, $^{177}$Lu-prostate-specific membrane antigen (PSMA), and PARPi (NCT03338790, NCT05682443, NCT02861573, NCT05733351) (Fig. 3A). Other trials using CTLA4 antibodies are being evaluated as monotherapy or in combination with PD-L1 inhibitors in advanced solid tumors, including mCRPC (NCT04896697). Results from the IMbassador250 trial established the importance of biomarker-driven trial design, where patients with high PD-L1 expression benefited the most from ICB (Powles et al, 2022). Investigation of avelumab in men with aggressive variant or NEPC showed no response except in one NEPC patient with high levels of natural killer (NK) cells, PD1+ T helper cells, and CXCR2+ T cells (Brown et al, 2022).

Multiple groups are investigating the use of immunocytokines to stimulate an anti-tumor response by activating T-cells. One trial is using a tumor-activated IL-12 fusion protein (XTX301) for solid tumors, including CRPC (NCT05684965). Another is leveraging the highly necrotic TME to deliver immuno-oncology agents. M9241 is a novel immunocytokine fusion of IL-12 and a DNA-histone binding protein, which accumulates in necrotic tumor tissue, leading to T-cell infiltration and activation (Xu et al, 2017). Combining M9241 with docetaxel has increased tumor-infiltrating lymphocytes and improved responses when paired with ICB in several cancers (Franks et al, 2023; Strauss et al, 2023). An ongoing trial is now testing the safety and efficacy of this combination in mCSPC and mCRPC (NCT04633252). Early results suggest the treatment is well-tolerated and shows promising response rates, offering hope for future clinical applications (Atiq et al, 2022). These studies highlight the importance of understanding the TME to identify patients who will most likely benefit from ICBs.

## CAR T cells

CAR T-cell therapy involves engineering a patient's T cells to express synthetic receptors that target a tumor-specific antigen. While there are no CAR T cells approved for PCa, several candidates are being explored. A phase I/II trial is testing STEAP1 CAR T in combination with ENZ in CRPC (NCT06236139), and preclinical studies suggest that the addition of IL-12 may further boost T-cell activation (Bhatia et al, 2023). In addition, another phase I trial is targeting prostate stem-cell antigen (PSCA)

## A. Immune checkpoint inhibitors

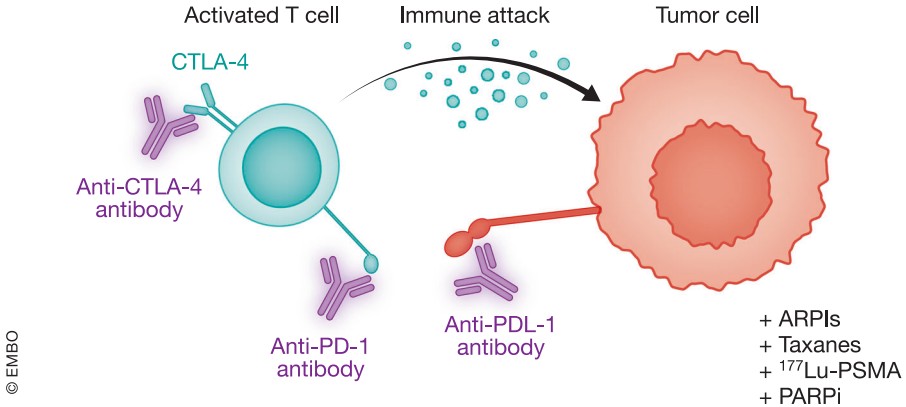

## B. Chimeric Antigen Receptor (CAR) T-cell therapy

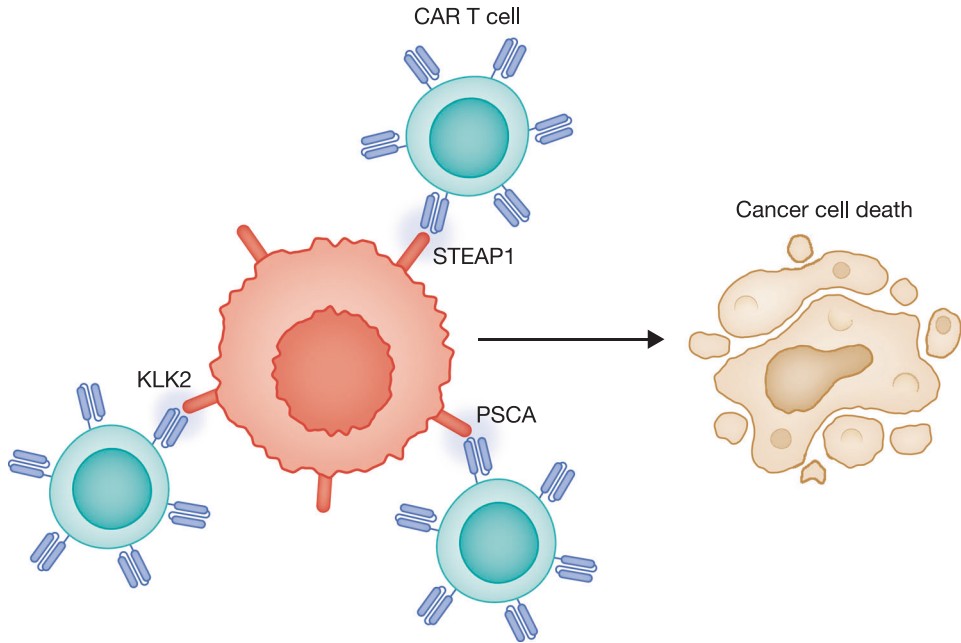

**Figure 3. Systematic immunotherapy in PCa.**

(**A**) ICBs such as anti-CTLA-4 and anti-PD-1/PD-L1 activate T-cells, enhance T cell-mediated immune responses, and promote immune attack against tumor cells. These therapies are currently being explored in combination with androgen receptor pathway inhibitors (ARPIs), taxane-based chemotherapies, $^{177}$Lu-PSMA, or PARP inhibitors (PARPi) to improve efficacy (NCT03338790, NCT05682443, NCT02861573, NCT05733351). (**B**) Chimeric Antigen Receptor (CAR) T-cell therapy involves engineering T cells to express CARs that recognize PCa-associated antigens such as KLK2, STEAP1, and PSCA. The binding of CAR T cells to these tumor antigens induces cancer cell death.

(NCT03873805), while another is targeting KLK2 (NCT05022849) in mCRPC (Fig. 3B). It is notable that, aside from exceptional responders, CAR T-cell therapy has not yet proven effective and remains limited by the risks of cytokine release syndrome and treatment-related mortality.

## Other immunomodulatory strategies

Different targets and approaches are being investigated to identify strategies that work for PCa. For example, a novel T-cell receptor agonist antibody that selectively activates and expands a subset of T cells is being evaluated for safety and tolerability in mCRPC (NCT05592626). Moving beyond biologics, other groups have developed a small-molecule inhibitor for Cbl-b, which acts as a negative regulator of T and NK cell activation, currently under clinical evaluation in mCRPC patients (NCT05107674).

PCa is considered a cold tumor, and immunotherapy in PCa has consistently faced challenges. This can be attributed to the lack of biomarker-driven trials for patient selection, poor understanding of the immune landscape, and their use as a

monotherapy. There has been a tremendous amount of work in the field of PCa immunotherapy with few if any tangible improvements, however, we believe that there are still many stones left unturned. Previous studies heavily focused on a narrow subset of immune cells, namely T-cells, but a more comprehensive approach will yield new avenues to target, such as NK-cells and macrophages. Furthermore, there needs to be more research to understand the TME and the unique immunological niche inside the prostate gland. We must also consider the impact of our standard of care treatments on the function of the immune system. Lastly, to improve response rates, we need to shift toward combination strategies and biomarker-driven therapies. Although the past trials have struggled, the insights gained remain valuable for refining treatment approaches and identifying synergistic drug combinations.

# Harnessing cell surface markers for precision therapy

Cell surface proteins are ideal therapeutic targets, enabling precision therapy. Their characterization has led to the development of innovative treatments like antibody–drug conjugates (ADCs), and radioligand therapies (RLTs), which use antibodies for tumor-specific antigens to deliver cytotoxic or radioactive payloads directly to tumors. Another class of cell surface-targeting drugs are bispecific T-cell Engagers (BiTEs), which redirect T-cells to attack tumor cells by simultaneously binding to a tumor-associated antigen (TAA) on cancer cells and CD3 on T-cells, inducing a direct cytotoxic response. Trispecific T-cell Activating Constructs (TriTACs) are a next-generation T-cell engager with improved half-life compared to BiTEs (Austin et al, 2021).

## PSMA

PSMA is highly expressed in AR-driven PCa but is minimally present in normal tissue. PSMA has been the primary focus of cell surface-targeting agents in PCa. Lutetium-177 labeled PSMA ($^{177}$Lu-PSMA), an RLT, has demonstrated remarkable success in treating PSMA-positive mCRPC (Wong et al, 2025). The VISION trial confirmed that adding Lu-PSMA to standard care improves PFS and OS, outperforming cabazitaxel by significantly reducing PSA levels and adverse events (Hofman et al, 2021; Sartor et al, 2021). In addition, its combination with $^{223}$Rd has shown encouraging results (Rahbar et al, 2023). $^{177}$Lu-PSMA was initially approved as a first-line monotherapy for PSMA-PET positive CRPC, but now, because of the PSMAfore trial (NCT04689828), its use has been expanded to include ARPI-treated mCRPC patients. This will add an additional layer to the treatment landscape of mCRPC and may delay the use of chemotherapies when appropriate. Multiple phase I trials are investigating next-generation PSMA RLTs (actinium-225-PSMA) in mCSPC and mCRPC (NCT06229366, NCT04597411), and in phase III trials in patients who have progressed on ARPI, taxanes, and $^{177}$Lu-PSMA (NCT06780670, NCT06402331).

PSMA has proven to be a robust target for PCa treatment, as a result, other cytotoxic modalities are being explored. For example, a phase I trial tested a PSMA-targeting ADC as monotherapy in progressive mCRPC and reported acceptable toxicity and anti-tumor activity (Petrylak et al, 2019). Another phase I trial is now testing it in combination with ARPIs in metastatic PCa (NCT04662580). In parallel, PSMA-targeting BiTEs are also under development (Chiu et al, 2020; Das et al, 2023; Leconet et al, 2018) and have shown anti-tumor activity in preclinical studies in mCRPC (Deegen et al, 2021). An ongoing clinical trial is evaluating PSMA-targeting BiTEs as monotherapy or in combination with PD1 inhibitors in advanced PCa (NCT05125016) (Fig. 4).

Despite promising outcomes, PSMA-targeted therapies face limitations. For example, treatment sequence seems to be important for PSMA-targeted ADC, since one study showed superior response when given before ARPI as opposed to after (Petrylak et al, 2020). Another study observed the potential of cross-resistance from prior treatment with PARPi in the case of $^{177}$Lu-PSMA (Raychaudhuri et al, 2025). Lastly, it is still unclear how the resistance mechanisms to PSMA-targeting ADCs and RLTs will manifest.

## Six-transmembrane epithelial antigen of the prostate 1 (STEAP1)

STEAP1 is minimally expressed in normal tissue, highly expressed in prostate adenocarcinoma (PRAD), and is absent in NEPC (Lee et al, 2018). A phase I clinical trial of a STEAP1 ADC failed due to dose-limiting toxicity in mCRPC (Danila et al, 2019). In contrast, a STEAP1-targeting BiTE has demonstrated early efficacy in mCRPC trials (Kelly et al, 2024) (Fig. 4).

## Tumor-associated calcium signal transducer 2 (TROP2)

TROP-2 has been identified as a potential therapeutic target and diagnostic marker for AR-driven mCRPC (Ajkunic et al, 2024; Sperger et al, 2023), with preclinical studies indicating effective imaging and localization of TROP2-positive tumors (Feng et al, 2024; Jiang et al, 2018). An ongoing clinical trial is evaluating a TROP-2 ADC in mCRPC patients who have progressed on ARPI (NCT03725761) (Fig. 4).

## B7-H3

B7-H3 (CD276) is an immunomodulatory protein highly expressed in CSPC and CRPC (Guo et al, 2023). A B7-H3 ADC is currently being evaluated in the neoadjuvant setting in intermediate and high-risk localized PCa (NCT02923180). Other ADCs are also being explored in advanced solid tumors, including mCRPC with bone-only disease (NCT05914116), and in mCRPC in combination with ARPIs (NCT06863272). Concurrently, a BiTE targeting B7-H3 is in a phase I trial as a monotherapy or in combination with an ICB in advanced solid tumors, including mCRPC (NCT05585034) (Fig. 4).

## Kallikrein-related peptidase 2 (KLK2)

KLK2 is among the most attractive targets for PRAD due to its prostate-specific expression profile (Hannu et al, 2014). Multiple phase I studies are exploring targeting it with a BiTE in advanced PCa patients (NCT04898634) or RLTs in mCSPC (NCT04644770) (Fig. 4).

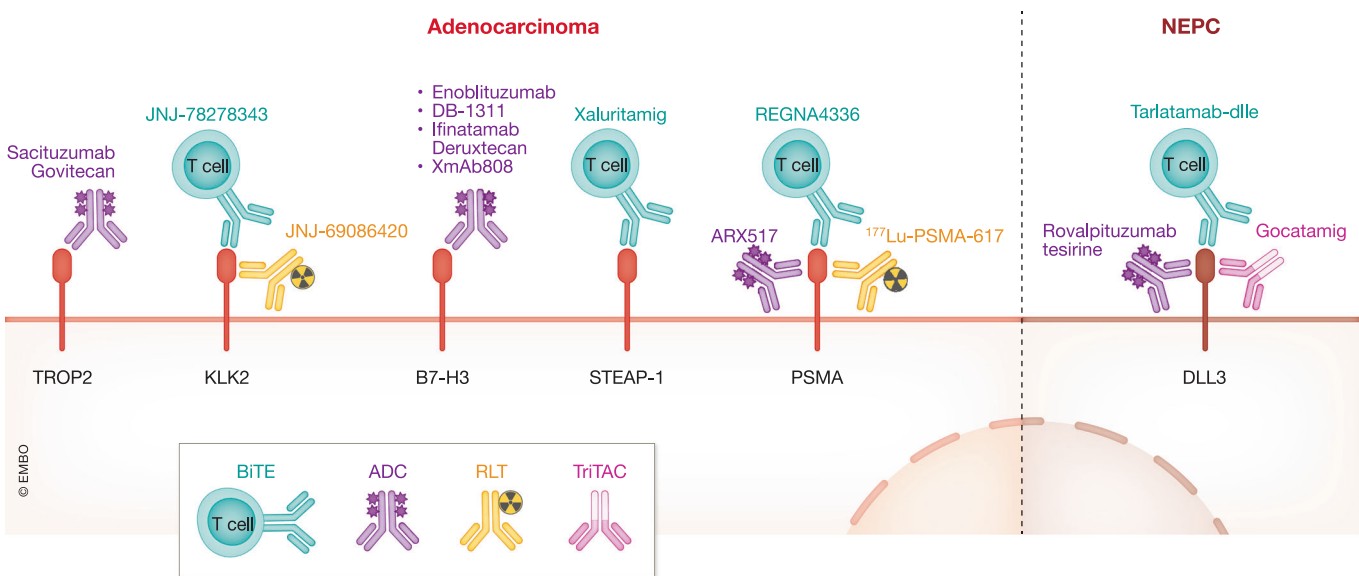

**Figure 4. Targeting the cell surface proteins in PCa subtypes.**

There are several therapeutic modalities to target the cell surface, such as antibody–drug conjugates (ADCs), radioligand therapies (RLTs), bispecific T-cell Engagers (BiTEs), and tri-specific T-cell activating constructs (TriTACs). Emerging ADCs, RLTs, and BiTEs are under clinical investigation for AR-dependent (left) and AR-independent (right) subtypes. Targets such as TROP2, KLK2, B7-H3, STEAP1, and PSMA are predominantly expressed in AR-driven tumors, while DLL3 is enriched in AR-independent, specifically NEPC. Representative agents include Sacituzumab Govitecan (TROP2-targeting ADC), JNJ-78278343 (KLK2 BiTE), JNJ-69086420 (KLK2 RLT), Enoblituzumab, DB-1311, Ifinatamab Deruztecan, and XmAb808 (B7-H3 ADCs), Xaluritamig (STEAP1 BiTE), ARX517 (PSMA ADC), REGN4336 (PSMA BiTE), $^{177}$Lu-PSMA-617 (PSMA RLT), Rovalipituzumab tesirine (DLL3 ADC), Gocatamig (DLL3 TriTAC) and Tarlatamab-dlle (DLL3 BiTE).

## Carcinoembryonic antigen-related cell adhesion molecule 5 (CEACAM5)

CEACAM5 is a promising NEPC cell surface marker, and preclinical studies supported the use of CEACAM5 ADCs in NEPC (DeLucia et al, 2021). Initially tested in colorectal cancer (Dotan et al, 2017), CEACAM5 ADC is now being explored for its applicability in managing NEPC (DeLucia et al, 2021), reflecting its potential utility across different tumor types with shared molecular markers.

## Delta-like canonical notch ligand 3 (DLL3)

DLL3 is another highly expressed cell surface marker in NEPC (Peddio et al, 2024). A DLL3 ADC is being evaluated in clinical trials in patients with solid tumors, including neuroendocrine carcinoma (Mansfield et al, 2021). A DLL3 BiTE (Tarlatamab-dlle) was recently approved for SCLC, and another BiTE (PT217) was recently granted fast-track designation by the FDA for neuroendocrine cancers, including NEPC. Another trial exploring a DLL3 TriTAC monotherapy as well as in combination with ICB or a B7-H3 ADC in advanced DLL3-expressing cancers, including NEPC, is currently in phase I (NCT04471727) (Fig. 4).

## Other cell surface targets

Additional cell surface targets are under investigation in PCa. For instance, EPCAM (CD326), previously identified as a prognostic biomarker in PCa (Liao et al, 2022), is now being investigated as a BiTE target in multiple advanced solid tumors, including PCa (NCT00635596). Moreover, TMEFF2 is an AR-regulated (Overcash

et al, 2013) PCa-specific transmembrane protein (Gery et al, 2002) that is being targeted with a BiTE in chemotherapy or ARPI-refractory mCRPC (NCT04898634). Lastly, a phase II trial targeting HER3 with an ADC in locally advanced or metastatic PCa is ongoing (NCT06172478).

Novel therapies under development that target cell surface markers promise to revolutionize the treatment landscape for PCa, mainly through advancing ADCs and RLTs (Table EV1). PCa phenotypes have distinct cell surface protein expression (Ajkunic et al, 2024; Sychev et al, 2024). For instance, AR-driven adenoCRPC is characterized by STEAP1, PSMA, TROP2, and B7-H3, while in NEPC highly expresses DLL3, somatostatin receptor (SSTR2), and CEACAM5. NEPC also shows high uptake of SSTR2 tracers. As a result, an ongoing clinical study is assessing the safety and efficacy of RLTs targeting PSMA, SSTR2, and gonadotropin-releasing hormone receptors (GRPR) in NEPC patients (NCT06379217). Research to identify other promising cell surface targets is ongoing; for example, an RLT targeting CDCP1 effectively inhibited DNPC growth in preclinical models (Zhao et al, 2022).

A limitation of ADCs/RLTs is that they are being developed to target individual tumor antigens and do not adequately consider tumor heterogeneity and multifocality. Therefore, the next frontier in cell surface-targeting therapies lies in a multi-faceted approach combining ADCs targeting multiple cell surface candidates and integrating biomarker-driven patient selection. A few trials are already incorporating this approach, such as co-targeting PSMA, SSTR2, and GRPR (NCT06379217) or DLL3 and B7-H3 (NCT04471727). This will provide a more robust and sustained anti-tumor effect as well as mitigate resistance mechanisms associated with single-target therapies by reducing the likelihood of tumor escape due to the loss or downregulation of a single target. Another key consideration is

incorporating diverse payloads that are appropriate for the disease. For example, a preclinical study identified a specific payload as highly potent in RB1-deficient tumors, despite low expression of the target antigen (Agarwal et al, 2023), suggesting that the choice of payload is as important as the selection of the target of interest.

BiTEs/TriTACs have their own set of limitations. For example, they may induce cytokine release syndrome, antigen loss resistance, immunosuppressive TME resistance, and limited T-cell infiltration into tumor sites (Zhou et al, 2021). Efforts are underway to improve BiTE by combining them with ICB or T-cell engagers, where they showed a potent anti-tumor response in mouse models (Moore et al, 2025) by amplifying their anti-tumor efficacy and overcoming resistance.

# Breaking the signal by targeting kinases

Kinase signaling pathways play a central role in several biological functions and contribute to lineage plasticity and treatment resistance in PCa, particularly by driving resistance to ARPIs through various mechanisms.

## IL-6/JAK/STAT pathway

Activation of the STAT3 pathway through the IL-6R or JAK1/2 activation has been shown to promote cancer stem cell (CSC)-like features and facilitate the transition from PRAD to NEPC (Chan et al, 2022; Deeble et al, 2001; Deng et al, 2022; Natani et al, 2022). Targeting upstream kinases like JAK1/2 with filgotinib/ruxlitinib (Chan et al, 2022; Deng et al, 2022) or neutralizing IL-6 ketotifen (Ji et al, 2023) has demonstrated potential in reducing stemness features in preclinical models, offering a possible therapeutic approach to prevent NEPC emergence.

## PI3K/AKT pathway

The PI3K/AKT pathway is frequently activated in advanced PCa, particularly in cases with PTEN loss (Wu and Huang, 2007). Its activation contributes to CSC characteristics and neuroendocrine differentiation (Bisson and Prowse, 2009), as well as treatment resistance (Toren and Zoubeidi, 2014), making it an attractive target for PCa (Bellezza et al, 2006). While early attempts to target this pathway with pan-PI3K inhibitors showed limited success (Cham et al, 2021), a phase III trial showed that the addition of Ipatasertib to ABI did not improve OS for men with mCRPC, regardless of PTEN status on IHC (de Bono et al, 2025). However, accurately defining PTEN status may be challenging due to variability in IHC cut-off thresholds and difficulty in calling PTEN copy number loss due to low cellularity. These technical and biological limitations highlight the need for more precise biomarker strategies when targeting the PI3K/AKT pathway. An ongoing phase 3 trial is evaluating capivasertib, a potent inhibitor of all three AKT isoforms (AKT1/2/3), in combination with ABI and ADT in mCSPC with PTEN loss (NCT04493853).

## TGF-β signaling

The TGF-β receptor and its co-receptor CD105 promote tumor progression in several cancers, including PCa (Thompson-Elliott et al, 2021). TGF-β signaling has been shown to promote AR activation and contribute to PCa development and progression (Shree et al, 2023). To test the hypothesis that TGF-β inhibition can promote sensitivity to ARPIs, a phase I trial is targeting CD105 along with APA compared to APA alone in mCRPC patients who progressed on ARPIs (NCT05534646).

## MAPK pathway

The MAPK pathway contributes to PCa progression, and it is activated downstream of receptor tyrosine kinases (RTK) (Zhang and Li, 2023). Targeting the RTK EGFR showed minimal activity as single agents in CSPC and CRPC (Canil et al, 2005; Pezaro et al, 2009; Small et al, 2007) while in combination with radiation showed reasonable tolerance and activity in non-metastatic PCa (Joensuu et al, 2010). MET is another RTK that activates MAPK, implicated in PCa progression (Zhang et al, 2018). A phase II trial demonstrated that targeting MET in mCRPC improved rPFS with mild toxicity (Monk et al, 2018). An ongoing trial is testing cabozantinib, an inhibitor targeting multiple RTKs (MET and VEGFR2), in combination with ICB in mCRPC (NCT05502315), building on evidence that MET inhibition may enhance anti-tumor immune responses (Jabbarzadeh Kaboli, Roozitalab et al, 2024). To enhance MAPK inhibition, a pan RTK inhibitor (MET, VEGFR2, AXL, and MER) was developed and is now in a phase I trial in mCRPC in combination with multiple ICBs (NCT05176483), though off-target effects and toxicity remain a concern.

## WNT and ROR2 signaling

Dysregulated WNT signaling is linked to AR-indifferent PCa, driving lineage plasticity and resistance (Bland et al, 2021; Koushyar et al, 2022; Tabrizian et al, 2023). Among its components, ROR2 has been identified as being highly expressed in NEPC and DNPC (Tabrizian et al, 2023). A ROR2-targeting ADC is under evaluation in squamous cell carcinoma (NCT05271604) and may hold promise for other squamous-like cancers such as DNPC. Furthermore, ROR2 is among the first cell surface markers to increase post-ARPI treatment, making it an attractive candidate to prevent disease progression (Tabrizian et al, 2023).

## Cyclin-dependent kinases

Cyclin-dependent kinases are a family of kinases that regulate various cell cycle checkpoints. CDK4/6 inactivates RB by phosphorylating it, resulting in E2F derepression and cell cycle progression. In PCa, CDK4/6 amplification has been linked to ARPI resistance (Han et al, 2017), and RB1 loss has been associated with lineage plasticity and NEPC. CDK4/6 inhibitors were tested in combination with ADT, but did not improve outcomes in RB-intact mCSPC (Palmbos et al, 2021); however, they showed efficacy in mCRPC in combination with docetaxel with prednisone (de Kouchkovsky et al, 2022). A preclinical study reported that CDK4/6 inhibition in combination with PARPi may induce apoptosis and halt NEPC differentiation (Wu et al, 2021), providing rationale for a clinical trial in NEPC. A trial assessing CDK9 inhibition in advanced solid tumor, including CRPC was recently completed, although results are yet to be published (NCT05159518). Other ongoing trials are exploring CDK4

(NCT04557449) and CDK7 (NCT05394103) inhibitors in advanced solid tumors, including PCa.

Despite the promise of kinase inhibitors in treating PCa, their limited clinical success highlights significant challenges. These include inadequate efficacy, off-target toxicity, activation of compensatory pathways, and the complexity of designing effective combination therapies. For example, AURKA/B inhibitors, designed to target Aurora A kinase and mitigate N-Myc overexpression in NEPC (Berger et al, 2019), have faced challenges in clinical translation due to resistance and complex protein interactions (Nikhil et al, 2020). Combination strategies involving kinase inhibitors are being explored to enhance treatment response, such as pan-AKTi with ARPI (NCT04493853) or pan-RTKi with ICB (NCT05176483). Many kinase inhibitor trials were not biomarker-driven, which raises the question of whether having a biomarker to select patients would have improved efficacy. Future research should focus on developing inhibitors with improved specificity, identifying biomarkers to predict response, and optimizing combination regimens. ctDNA has previously been used to identify patients who may benefit from PI3K inhibition (Crabb et al, 2021; Sweeney et al, 2021), and a similar strategy could be implemented for other kinase inhibitors.

# Targeting the epigenome as a therapeutic frontier

The distinct phenotype and transcriptional reprogramming observed in AR-independent compared to AR-driven CRPC (Beltran et al, 2016; Labrecque et al, 2019) cannot be entirely attributed to the limited genomic variations between these two phenotypes, suggesting that progression to AR-independence is driven by epigenetic dysregulation (Chakraborty et al, 2023; Cheng and Wang, 2021; Davies et al, 2021a, Davies et al, 2023; Davies et al, 2020; Nouruzi, Ganguli et al, 2022). Considerable research focusing on epigenetic re-programming underlying the emergence of lineage plasticity (Chakraborty et al, 2023; Ge et al, 2020) may enable reversion of these phenotypes to a luminal state, delay of this transition, or identify potential therapeutic vulnerabilities. Epigenetic regulation occurs at multiple levels, including chromatin organization, DNA methylation, and histone modification. Each of these modifications presents potential targets to assess and exploit PCa vulnerabilities.

## Histone modifications as therapeutic frontiers

### Epigenetic writers
Epigenetic writers deposit post-translational modifications onto histones, such as lysine methyltransferases and histone acetyltransferases (HATs).

EZH2, the catalytic subunit of the PRC2 complex, deposits methyl groups onto lysine 27 of histone 3 (H3K27). EZH2 plays diverse roles in PCa (Nouruzi et al, 2023; Park et al, 2021), such as silencing the luminal program (Davies et al, 2021a) and suppressing canonical AR signaling (Dardenne, Beltran et al, 2016). EZH2 is overexpressed and post-translationally modified in NEPC compared to PRAD (Davies et al, 2021a, Duan et al, 2020; Nouruzi et al, 2022; Nouruzi et al, 2023). Depending on the PCa stage under treatment pressure, EZH2 inhibition can reverse lineage plasticity

when cells are in a metastable state post ARPI to a luminal-driven state (Davies et al, 2021a), or push NEPC cells toward a highly differentiated state with increased NE programs (Venkadakrishnan et al, 2024). Although EZH2 inhibitors (EZH2i) have shown promise in preclinical studies (Gulati et al, 2018; McCabe et al, 2012; Miranda et al, 2009; Verma et al, 2012), they have had modest anticancer activity as a monotherapy in PCa (Yap et al, 2019). However, EZH2i in combination with ARPI showed efficacy in preclinical studies (Davie et al, 2021b, Fischetti et al, 2024) and in mCRPC (NCT03460977), which supports the phase 3 trial in mCRPC progressed on Abi (NCT06551324). Moreover, targeting dual EZH1/2 inhibitors is under investigation in metastatic PCa (NCT06632977). Resistance to EZH2 inhibitors remains challenging due to mutations in EZH2 and the activation of alternative pathways (Nouruzi et al, 2023; Kaur et al, 2024). In addition, EZH2 regulates broad cellular functions, raising concerns about off-target toxicity [19].

The histone acetylation profiles of PRAD and NEPC are distinct (Baca et al, 2023; Baca et al, 2021), highlighting the importance of this histone modification and HATs in disease progression. While there are no clinically approved HAT inhibitors, a phase I trial of the KAT6 inhibitor PF-07248144 in solid tumors (NCT04606446) has shown durable antitumour activity in breast cancer (Mukohara et al, 2024). Whether it has potential as a therapeutic agent for mCRPC remains to be investigated.

### Targeting epigenetic erasers
LSD1 (KDM1A) demethylates H3K4 and H3K9, suppressing AR-driven CRPC-associated genes, upregulating neuroendocrine programs (Coleman et al, 2020; Li et al, 2020), and repressing TP53 (Kumaraswamy et al, 2023). Preclinical studies showed that the LSD1 inhibitor has activity in both activities in AR-driven CRPC (Li et al, 2023) and NEPC (Jasmine et al, 2024; Kumaraswamy et al, 2023). A dual LSD1/HDAC6 inhibitor, JBI-802, has shown tolerability and efficacy in Myeloma (Naveen Sadhu et al, 2021), and it is currently in phase I/II trials in patients with advanced solid tumors, including NEPC (NCT05268666).

### Epigenetic readers
BET proteins, including BRD2, BRD3, BRD4, and BRDT, are epigenetic readers that bind to acetylated lysine residues on histones, resulting in transcriptional activation to sustain oncogene expression and tumor growth (Filippakopoulos and Knapp, 2014). BRD4 interacts with AR in CRPC (Asangani et al, 2014), supports oncogene activation (Shafran et al, 2021, Zhang et al, 2025), and contributes to treatment resistance (Kim et al, 2021). BET inhibitors have effectively disrupted transcriptional programs driven by oncogenic transcription factors (TFs) (Kim et al, 2021; Shafran et al, 2021; Wyce et al, 2013) and inhibit tumor growth and metastasis (Delmore et al, 2011; Faivre et al, 2017; Shafran et al, 2021; Wyce et al, 2013; Xiang et al, 2018). However, the clinical translation of BETi as a monotherapy has faced some challenges. For example, SPOP mutations, which are common in primary PCa, predict strong responses to ARPIs but confer resistance to BETi (Abida et al, 2019b, Boysen et al, 2018; Swami, Graf et al, 2022b, Zhang et al, 2017), emphasizing the importance of genomic profiling to identify candidate patients and using BETi in combination with other agents. Currently, BETi is under evaluation in a phase II trial in combination with ENZ in mCRPC

(NCT04986423). Other ongoing trials are investigating it in combination with a CBP/p300 (HAT) inhibitor in advanced solid tumors, including CRPC (NCT05488548), as well as ARPI and ICB in mCRPC (NCT04471974).

### DNA methylation as therapeutic frontiers

DNA methylation, catalyzed by DNA methyltransferases (DNMTs), involves the addition of a methyl group to the cytosine residue to form 5-methylcytosine (5mC) (Lyko, 2018). Conversely, DNA demethylation is mediated by the TET family (Joshi et al, 2022; Kohli and Zhang, 2013). DNA methylation distinguishes between PRAD and NEPC, suggesting the potential of targeting DNA methylation (Farah et al, 2022; Gravina et al, 2013; Yamada et al, 2023). Targeting DNMTs restores sensitivity to ARPIs in preclinical models (Farah et al, 2022), which paved the way for a phase I trial investigating DNMT inhibition (DNMTi) in combination with ENZ (NCT05037500). In RB1-deficient CRPC and NEPC, DNMTi leads to increased expression of B7-H3, and using B7-H3 targeted ADC in combination with DNMTs leads to tumor regression in preclinical models (Yamada et al, 2023). Similarly, PSMA is suppressed in NEPC through the gain of DNA methylation and loss of H3K27ac at its locus (Sayar et al, 2023), suggesting that HDAC and DNMT inhibitors may restore PSMA expression.

Complementary to the role of DNMTs, TET enzymes have been implicated in maintaining lineage fidelity and mitigating therapy-induced plasticity (Xu et al, 2024), presenting another therapeutic avenue. Although preclinical studies suggest that TET2 inhibition can prevent the progression of lineage plasticity (Xu et al, 2024), more research needs to be done before TET inhibitors enter clinical trials in PCa.

### Chromatin dynamics as therapeutic frontiers

Chromatin remodelers modify nucleosome positioning, density, and accessibility to enable or restrict the binding of transcription factors, DNA repair processes, DNA replication, and transcriptional machinery. In PCa, chromatin remodeling complexes such as SWI/SNF and CHD have been linked to lineage plasticity and promote oncogenic transcriptional programs (Augello et al, 2019; Cyrta et al, 2020).

The SWI/SNF complex is a key chromatin remodeling complex in PCa and is required for maintaining chromatin accessibility at AR sites (Xiao et al, 2022). One of the most frequently altered SWI/SNF members is ARID1A, and recent preclinical studies have shown that tumors with ARID1A mutations tend to respond better to ICB (Zhou et al, 2024). An ongoing clinical trial investigates the combination of PD1 and LAG3 inhibition in metastatic or unresectable solid tumors with ARID1A mutations, including PCa (NCT04957615). While ARID1A mutations are infrequent in PCa, studies have shown that low ARID1A expression is associated with adverse disease outcomes, including increased inflammation and immune suppression (Li et al, 2022), suggesting that perhaps patients with low ARID1A expression may benefit from ICB. SMARCA2/4 are the enzymatic subunits of the SWI/SNF complex, and inhibitors targeting them are under preclinical investigation (Vaswani et al, 2025; Xiao et al, 2022). While there haven't been any trials for SMARCA2/4 inhibitors in PCa, a phase I trial for an inhibitor in AML is ongoing (NCT04891757) and may be readily translated to PCa if successful.

While some epigenetic drugs have successfully progressed to clinical use (Fig. 5; Table EV2), many others face limitations due to resistance mechanisms, off-target effects, and toxicity. These challenges have led to skepticism regarding epigenetic target therapeutic potentials. However, the dynamic and evolving nature of PCa provides compelling reasons to persist in investigating epigenetic therapies. The increasing prevalence of AR-independent subtypes offers an opportunity to re-evaluate the efficacy of epigenetic inhibitors that may have shown limited success in earlier trials. Previously, a lack of patient stratification by molecular subtype likely masked the potential benefits of these therapies for AR-independent populations. DNA methylation marks on ctDNA can be used to distinguish PCa phenotypes (Beltran et al, 2016; Beltran et al, 2020) and identify potential candidates for combination therapies such as DNMTi with B7-H3 or PSMA ADCs. Furthermore, ctDNA fragmentation profiles could be used to quantify the proportion of PRAD and NEPC in patient tumors (De Sarkar et al, 2023). However, it is important to note ctDNA alterations may not reflect the true transcriptomic state of the tumor, and other complementary methods should be employed concurrently.

A major limitation of initial trials of epigenetic inhibitors was their application as monotherapies. Growing evidence suggests that combination regimens may better exploit lineage-specific vulnerabilities. For instance, targeting EZH2 in combination with ARPIs in a meta-plastic state could simultaneously address the epigenetic mechanisms driving lineage plasticity, restore AR sensitivity, and inhibit AR post-activation. Similarly, HDAC or DNMT inhibitors may resensitize tumors to ARPIs. Epigenetic drugs may also sensitize tumors to ICB by inducing viral mimicry and triggering immune activation (Morel et al, 2021). Therefore, epigenetic drugs may be combined with a wide array of agents to overcome the limitations observed so far and provide a new paradigm for treating advanced PCa.

# Metabolic architecture of prostate cancer: opportunities for intervention

Metabolic reprogramming, a hallmark of cancer (Hanahan, 2022), drives PCa progression through altered lipid (Chen et al, 2024; Lounis et al, 2020; Swinnen et al, 1997; Wu et al, 2014), amino acid (Chen et al, 2024; Strmiska et al, 2019), and energy metabolism (Crowell et al, 2023; Pujana-Vaquerizo et al, 2024; Varuzhanyan et al, 2024). Localized PCa is highly dependent on lipolysis (Mah et al, 2020) but shifts toward glycolysis (Wang et al, 2020; White et al, 2018) and choline metabolism (Beier et al, 2023) in CRPC (Bader and McGuire, 2020). In AR-independent states, such as those in NEPC, TFs like SOX2 (de Wet et al, 2022) and MYC (Crowell et al, 2023) mediate metabolic reprogramming to sustain growth and therapy resistance. These metabolic changes support tumor survival, provide building blocks for epigenetic modifications, and contribute to therapy resistance. Metabolic therapies aim to exploit these metabolic vulnerabilities by targeting specific metabolites or metabolic pathways that cancer relies on.

## Lipid metabolism

Lipid metabolism is a hallmark of PCa metabolism. Fatty acid synthase (FASN) inhibitors and cholesterol-lowering agents are being evaluated in multiple clinical trials. A Phase II trial is

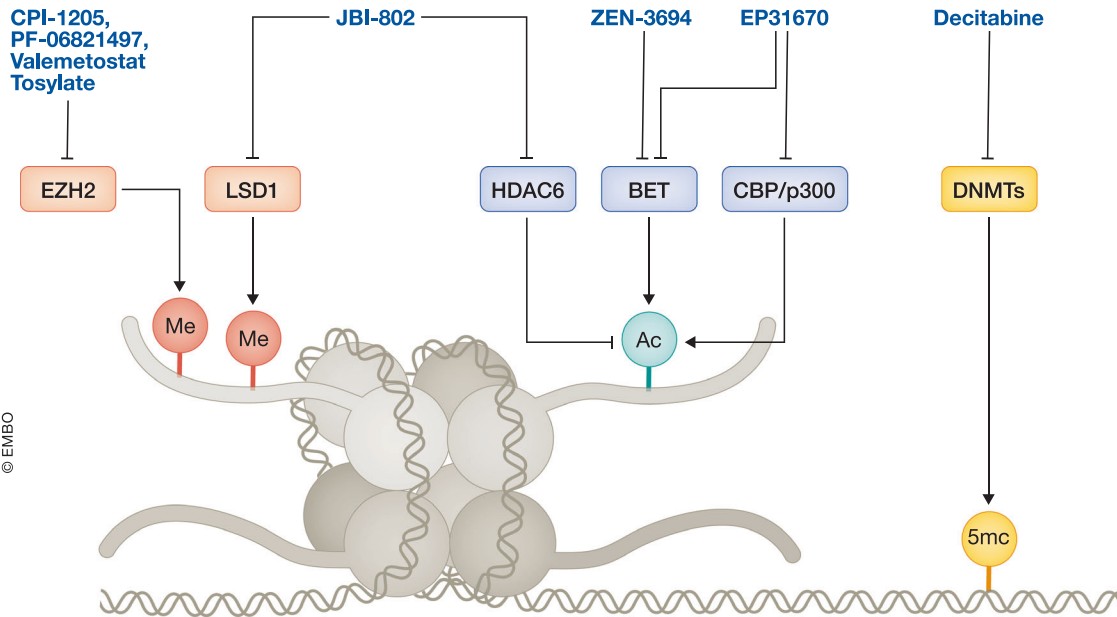

**Figure 5. Epigenetic regulators as therapeutic targets.**

Several inhibitors are in clinical trials for targeting epigenetic machinery, including EZH2 (CPI-1205, PF-06821497, and Valemetostat Tosylate) and LSD1 (JBI-802); histone deacetyl transferase HDAC6 (JBI-802); histone acetylation readers BET (ZEN-3694, and EP31670), histone acetyltransferases CBP/p300 (EP31670); as well as DNA methyltransferases (DNMTs) (Decitabine).

investigating FASN inhibition in combination with cabazitaxel in docetaxel-resistant PCa and CRPC (NCT04337580), while another inhibitor is being evaluated in combination with ENZ (NCT05743621). In parallel, the Phase III trial is evaluating whether atorvastatin, a cholesterol-lowering agent, in combination with ADT can delay castration-resistance compared to ADT alone (NCT04026230).

## Glycolysis

The Warburg effect, where cancer cells preferentially use aerobic glycolysis over oxidative phosphorylation (Liberti and Locasale, 2016), is prevalent in advanced PCa, particularly in PTEN-deficient cases (Courtnay et al, 2015). Glycolysis inhibitors, such as 2-deoxyglucose, have shown acceptable toxicity profiles in patients (Raez et al, 2013; Stein et al, 2010). Other trials targeting glycolysis using lonidamine, a hexokinase inhibitor that disrupts glycolysis, showed promise for treating benign prostatic hyperplasia (BPH) (Ditonno et al, 2005) but no efficacy in CRPC patients (Boccardo et al, 1992). A novel small molecule with antiglycolytic activity has demonstrated preclinical efficacy in inhibiting glycolysis in advanced PCa xenografts, resulting in growth suppression equivalent to ENZ (Uo et al, 2024), suggesting the potential of translating this combination to the clinic.

Lactate production is a downstream metabolite of aerobic glycolysis and is important for multiple oncogenic processes. Increased lactate is a key metabolic signature of NEPC (Choi et al, 2018). This observation provided a rationale for targeting the highly expressed lactate transporter MCT4, which has shown efficacy in preclinical AR-independent models (Choi et al, 2016; Choi et al, 2018; Sun et al, 2019).

## Amino acid metabolism

Glutamine is a key amino acid supporting PCa progression (Erb et al, 2024; Marin-Aguilera et al, 2021). Glutaminase-1 (GLS1) metabolizes glutamine to glutamate, supports tumor survival and invasiveness (Pan et al, 2015). GLS1 inhibitors have shown promise in patients with advanced solid tumors, reporting encouraging results in PCa (Harding et al, 2015). A Phase II trial is evaluating its efficacy in combination with PARPi in HRR-proficient CRPC (NCT04824937), however, the status of the trial is uncertain.

A frequently observed metabolic disruption in cancer is the decreased ability to synthesize arginine due to ASS1 suppression (Zou et al, 2019). This disruption allows tumor cells to divert aspartate toward pyrimidine synthesis to meet the high nucleotide demands (Rabinovich et al, 2015). However, this metabolic reprogramming leaves tumor cells highly dependent on exogenous arginine for survival. ADI-PEG20 was developed to target this vulnerability by stabilizing arginine deaminase, leading to arginine deprivation and cell death. A phase I trial has shown acceptable safety results in solid tumors (Tomlinson et al, 2015) and is now in phase II in combination with carboplatin and cabazitaxel in aggressive variant PCa (NCT06085729).

PCa progression is characterized by metabolomic dysregulation. An interesting avenue to target the tumor metabolome is through dietary intervention. The importance of diet lies not only in its general health benefits but also in its direct influence on the metabolome, which in turn shapes the epigenome and transcriptome. Several clinical trials have explored the role of certain dietary factors in PCa development and progression, such as vitamin E and K supplementation (Klein et al, 2011; Swamynathan et al, 2024), sulforaphane (Traka et al, 2019), phospholipids (Brasky et al, 2013),

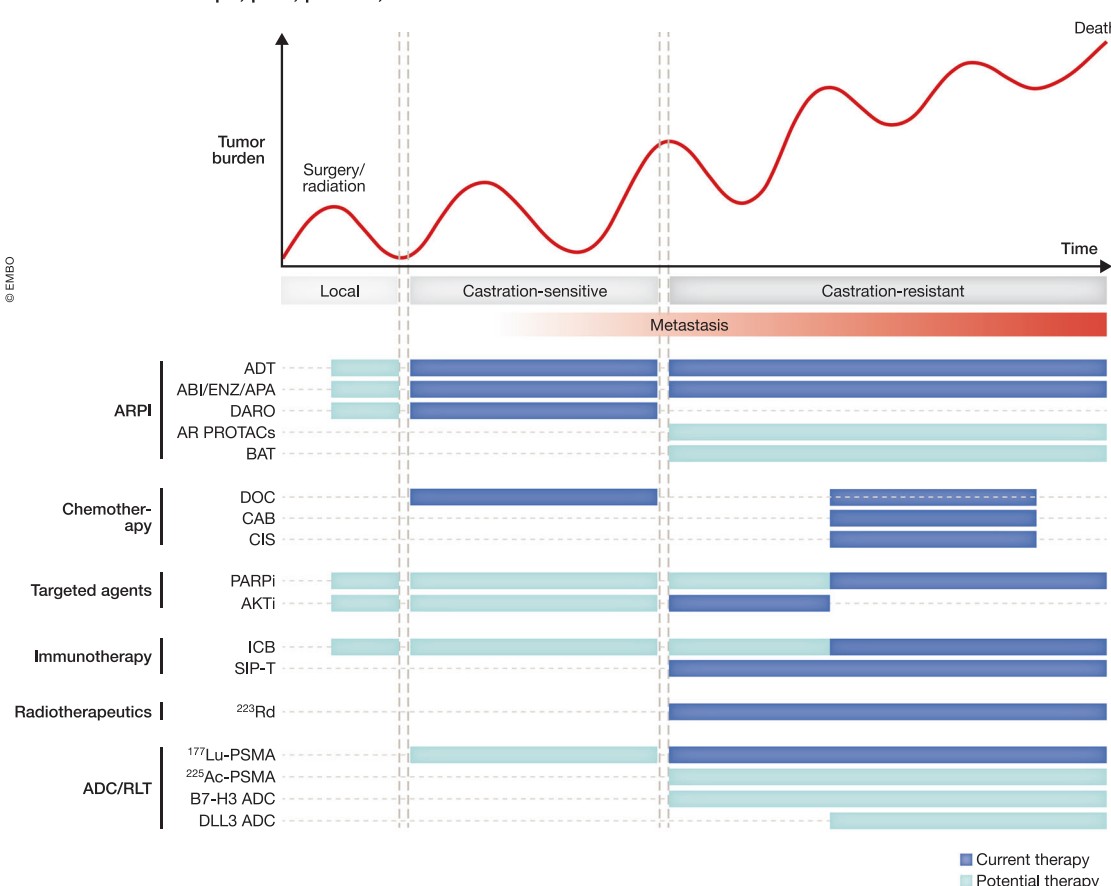

**Figure 6. The evolving landscape of PCa treatment.**

A schematic representation of prostate cancer therapies, highlighting key milestones in treatment development, emerging resistance mechanisms, and future directions.

and others (Table EV3). These studies emphasize the potential of diet-based strategies to target metabolic vulnerabilities and improve patient outcomes.

Metabolic changes can influence the epigenomic landscape. For example, increased serine and one-carbon biosynthesis have been linked to promoting lineage plasticity (Reina-Campos et al, 2019), while decreased MAT2A activity, the enzyme converting methionine to SAM, can lead to a hypomethylated DNA landscape in DNPC (Li et al, 2025). PCa phenotypes are marked by unique transcriptomic and epigenomic profiles. It is reasonable to assume different phenotypes will have distinct metabolic requirements, tailored to support their specific characteristics. Future research should aim to decode how the metabolome sustains cancer phenotypes and explore how dietary interventions or pharmacological strategies can exploit these dependencies. However, selectively disrupting tumor metabolism without harming healthy tissues is challenging.

## Future directions and clinical implications

Targeting AR has long been the cornerstone of PCa therapy, and remains the primary target through decades of research and clinical

advancements. This focus is justified, as AR is a central driver of PCa progression, and its inhibition has yielded significant clinical benefits. Most of the advancements in PCa therapeutics have been towards targeting the AR itself or downstream targets of the AR. In light of challenges such as the rising incidence of ARPI resistance and lineage plasticity, the question arises whether we are doing enough to address the evolving complexity of advanced PCa. Diagnosis of AR-independent subtypes remains a challenge, as a definitive histopathological diagnosis from biopsies is often not feasible and may fail to capture the tumor's heterogeneity. This highlights the urgent need for precise diagnostic tools to classify tumors accurately, segregate patients effectively, and monitor treatment responses or early resistance (Okasho et al, 2021). Perhaps an explanation for the lack of progress in non-AR targeting agents in PCa is due to inadequate patient stratification enrolling in those trials and overlooking the heterogeneous nature of the disease. To maximize the likelihood of positive trials, widespread integration of genomic, transcriptomic, epigenetic, along histological classifiers will help in treatment planning and management.

As the treatment landscape of PCa continues to evolve, we are likely to see a shift in therapeutic strategies that reflects the profound understanding of disease biology. We speculate that novel combinations, such as PARPi and ARPI or Lu-PSMA, to move

earlier in the treatment paradigm, potentially offering curative-intent strategies for high-risk localized PCa. The approval of PSMA-targeted RLT will pave the way for other ADCs, including B7-H3, TROP2, and STEAP1, in PRAD. It is reasonable to assume that ADCs targeting DLL3 will likely become a standard of care for NEPC, given that one was recently approved for SCLC. Lastly, despite challenges, immunotherapy is still being explored as a therapeutic strategy in PCa. For example, molecular stratification and understanding of disease biology have paved the way for the approval of Pembrolizumab in MSI-hi patients. Further research will identify other candidates for ICB as monotherapy or in combination with other agents (Fig. 6). Finally, it is important to stress the need for recognizing inter- and intra-tumor heterogeneity, biomarker-driven patient selection, and rationale combination therapy for future clinical trials. We remain hopeful that combinatorial strategies will improve the survival of PCa patients.

## Pending issues

Novel agents targeting diverse aspects of PCa biology are under active development and ae being tested in clinical trials. Nevertheless, the success rate of these trials remains low due to several reasons. Integrative diagnostic approaches that combine histopathology with genomic, transcriptomic, and epigenomic profiling are not yet widely implemented in clinical settings. The lack of validated biomarkers to reliably classify PCa subtypes continues to limit effective patient stratification. In addition, tumor heterogeneity and an incomplete understanding of the underlying biology of distinct subtypes further complicate therapeutic decision-making.

Combination therapies are being adapted to overcome these challenges; however, there are several key considerations required to optimize treatment, such as timing, sequence, and patient selection. Efforts to harness immunotherapy in PCa have thus far yielded limited results, but it may be premature to dismiss its potential beyond rare genomic contexts. A deeper understanding of the tumor microenvironment and immune landscape could unlock new opportunities. Lastly, emerging areas such as metabolomics remain underexplored and may reveal novel therapeutic vulnerabilities worthy of clinical pursuit.

## Peer review information

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

## Acknowledgements

We thank all the members of the Zoubeidi laboratory for their valuable input in conceptualizing and progressing this review article. We would like to thank the funding agencies. This research was supported by funding from the Terry Fox Research Institute New Frontiers Program (to AZ), the Canadian Institutes of Health Research (195886; to AZ), the Prostate Cancer Foundation Young Investigator Award (to SN), and the US Department of Defence Early Investigator Award (PC230507, to SN).

## Author contributions

**Shaghayegh Nouruzi**: Conceptualization; Data curation; Formal analysis; Visualization; Writing—original draft; Writing—review and editing. **Maxim Kobelev**: Conceptualization; Data curation; Formal analysis; Funding acquisition; Visualization; Writing—original draft; Writing—review and editing. **Nakisa Tabrizian**: Conceptualization; Data curation; Formal analysis; Visualization; Writing—original draft; Writing—review and editing. **Martin Gleave**: Supervision; Writing—review and editing. **Amina Zoubeidi**: Conceptualization; Supervision; Funding acquisition; Writing—review and editing.

## Disclosure and competing interests statement

Martin Gleave is on the editorial advisory board. This has no bearing on the editorial consideration of this article for publication.

