## [Peer Review File · EMBO Molecular Medicine]

New frontiers in prostate cancer treatment from systemic therapy to targeted therapy

Shaghayegh Nouruzi, Maxim Kobelev, Nakisa Tabrizian, Martin Gleave, and Amina Zoubeidi

Corresponding author(s): Amina Zoubeidi (azoubeidi@prostatecentre.com)

Review Timeline:

Submission Date:	7th May 25
Editorial Decision:	23rd May 25
Revision Received:	11th Jun 25
Accepted:	16th Jun 25

Editor: Lise Roth

Transaction Report:

23rd May 2025

Dear Amina,

Thank you for the submission of your review to EMBO Molecular Medicine. We have now received feedback from the experts who agreed to evaluate your manuscript.

As you will see from the reports below, they found the review overall well-written, new and interesting, and are supportive of publication pending minor revisions.

We will therefore welcome the submission of a revised manuscript addressing the referees' comments and suggestions. As suggested by referee #3, the various "Future perspectives" sections could be integrated into a single section at the end of the review.

Please also address the following editorial concerns:

- We can accommodate of 5 keywords, please adjust accordingly.
- Please merge Funding with Acknowledgements.
- Please add a Disclosure and competing interests statement. Please review the policy <https://www.embopress.org/competing-interests> and update your competing interests if necessary. Please add: "Martin Gleave is an editorial advisory board. This has no bearing on the editorial consideration of this article for publication."
- CRediT has replaced the traditional author contributions section because it offers a systematic machine readable author contributions format that allows for more effective research assessment. Please remove the Authors Contributions from the manuscript and use the free text boxes beneath each contributing author's name in our system to add specific details on the author's contribution.
- References should be listed in alphabetical order, with 10 authors before et al.
- The manuscript should contain a Glossary explaining some of the terms used for laymen. Glossary is not a list of abbreviations; abbreviations should be defined the first time they are used in the text.
- The manuscript should also contain a section termed "Pending issues": At the end of each article, there is a box highlighting issues that still need further studies and where research efforts should converge (called the Pending issues box). This should be limited to a few points and should be shorter than the "Future perspectives" paragraphs.
- The figure legends should be moved to the end of the manuscript text.
- Suppl. tables should be renamed Table EV1 - EV3 and all need legends added to the top of the page.
- Table legends and duplicate figure legends should be removed from the manuscript text.
- The figures should be removed from the manuscript text and should be uploaded as individual files in EPS, TIFF or PDF format. Fig 5 is missing a legend.
- Please also note the following points:
 1. We work with one of our expert scientific illustrators, who will assist with getting the figure to a publication ready state. What we need from you is a draft that accurately illustrates the key scientific concepts that you wish to show. Please also ensure that the figure draft is conceptually as close as possible to the final version as we cannot offer to do multiple rounds of revisions (and substantial changes might necessitate a complete redesign).
 2. If there are certain aspects of your figures that are based upon assumptions or where the scientific data remains ambiguous, please add a comment so that we can work with you on an accurate depiction. Please ensure the directionality and nature of interactions is presented accurately.
 3. If the figure or single panels of the figure have been adapted from a published figure, please add this information to the figure legend (e.g., 'Adapted from...' or 'Based on...').
 4. Please only re-use figures or parts of a figure if this is essential for understanding the concept communicated. If the figure contains re-used images or elements of images, please make sure that you have the permission/license to publish it. All re-used material must be explicitly cited.
 5. If you use an image data base for scientific iconography (e.g., BioRender), please let us know if you have a license that allows for publication in an academic journal.

Looking forward to receiving your revised manuscript,

With my best wishes,

Lise

Lise Roth, Ph.D
Senior Editor

***** Reviewer's comments *****

Referee #1 (Remarks for Author):

The review by Nouruzi and colleagues is comprehensive, timely, and powerful. It covers a large body of work in a compelling narrative, and not only includes a full list of supporting publications but also complements this with a full list of clinical trials and therefore the review is exhaustive. My comments are largely clarifications and some organizational queries.

Comments:

- 1 Add the rationale for developing AR degraders. Is there an anticipated benefit for degrading the AR compared to antagonizing the receptor? Is that going to be an issue - what was set back in EPI-7386? Wouldn't just get AR independent aggressive lineage? Is there a risk of rapidly developing AR independent aggressive lineages?. What was set back in EPI-7386?
2. Given the AR is a pro-differentiation factor, why is the BAT trial counterintuitive?
3. Future Challenges on AR therapies - perhaps also mention fragmentomic approaches to develop liquid biopsy that will give better/comprehensive evaluations of AR targeted therapies (i.e. monitor which enhancers are being targeted)
4. In discussing the identification of actionable genomic alterations it might be useful to highlight the challenges of developing comprehensive catalogs (tissue access, timing etc)
5. On the section of PARPi sensitivity there seems to be a disconnect between tumors that are sensitive and the levels of key mutations? Is this the case? Does the concept of "BRCAness" explain how much sensitivity occurs independent of altered BRCA1/2 mutation status for example? Is that a concept?
6. Why is it surprising that TRIUMPH was null? Clarify why PARPi without ADT had no benefit - is that added? Or is there pathway convergences? Is there a rationale explanation? Likewise can the authors speculate on what the functional assay would be used to stratify HRR activity more precisely/
7. Are there molecular profiles that capture high MSI patients? Are they consistent with other tumors where ICB is effective? Are there lessons to be drawn?
8. Add a line to explain what CDK12 is and why it links to HR deficiency
9. More generally, what are the top muts/alterations in advanced PCa, and how many are currently being targeted? (add to F1? Or table?)
10. The section of immune-based therapies is interesting, but what comes through is the lack of success in this arena. In the future perspective perhaps the authors could also add some sense of at what point does it appear that the scientific community have undertaken sufficient work to accept that they cannot reject the null hypothesis, ie that PCa is not sensitive to immunotherapy? Or what would that debate look like? What would be the burden of endpoints?
11. What is rationale for order of sections (# of trials? # approved drugs/strategies? Etc?). An overall figure showing all the different strategies would help the reader (like the last figure, but as a cell-based cartoon?)
12. For the cell surface-based approaches (e.g. ADC) is there any sense for when a marker is targeted successfully (perhaps PSMA) and when not - (perhaps STEAP1) - more generally, is it type of marker - how much is external and how cycled? Docking studies? Any rationale clues?
13. Combinatorial ADC/RLTs - how decide which combination? Or is it investigator-based rationale?
14. P.12 - MAPK pathway - the (REF) needs to be replaced and (329133968) in CDK section
15. The epigenetic section (perhaps epigenomic?) after AR may make more sense?

16. In the epigenetic section, perhaps clarify in what ways histone states and DNA methyl differ between PRAD and NEPC (where, gain, loss, etc) and illustrate how that contributes to differences?

Referee #2 (Remarks for Author):

Nouruzi et al. discuss the main treatment options for prostate cancer (PC) across its different stages, ranging from localized disease to neuroendocrine prostate cancer (NEPC) and double-negative prostate cancer (DNPC). The manuscript primarily focuses on therapies evaluated in clinical trials, while also including some that have shown promising results in preclinical studies. Throughout the article, the authors emphasize the importance of personalized treatment approaches considering the disease's heterogeneity. Notably, the manuscript goes beyond a simple summary of the literature and provides expert insights into the key questions and challenges for the future of the field.

I have a few suggestions to further strengthen the manuscript:

1. Define androgen deprivation therapy (ADT), as this will benefit readers who are not specialists in prostate cancer.
2. List all authors in the bibliography to help readers identify the research groups and laboratories.
3. Because there is an extensive literature on preclinical studies in PC, it would be helpful if the authors specify in the review the criteria that they used for selecting the preclinical studies that were included.
4. Consider including a section on stereotactic body radiation therapy (SBRT) and cite recent relevant literature, such as: Nikitas J. *JAMA Oncol.* 2025 May 15:e251059. doi: 10.1001/jamaoncol.2025.1059.

Referee #3 (Remarks for Author):

This is a timely, well-structured, and deeply researched review which appears to be within the scientific scope of your journal. The manuscript provides valuable insights into prostate cancer (PCa) therapy, from its molecular mechanisms to clinical applications.

The article provides a thorough and structured overview of the evolving PCa treatment landscape, spanning conventional systemic therapies (ARPI, chemotherapy, radiotherapy) to emerging targeted strategies (ADCs, RLTs, immunotherapy, metabolic and epigenetic approaches). It also nicely captures the complexity of AR-driven and AR-independent subtypes (e.g., NEPC, DNPC), and articulates resistance mechanisms and lineage plasticity.

The review effectively links biomarkers to clinical applications, thus underscoring the translational significance of genetic stratification and ctDNA analysis. The discussion of ctDNA-based monitoring is appropriately critical. The inclusion of ongoing clinical trials (with trial IDs and disease settings) enhances its translational depth. Tables and figures summarising trial data and therapeutic mechanisms are helpful for both researchers and clinicians working in this field. Importantly, the review covers not only what is approved, but also what is investigational, clearly positioning the trajectory of this field. Lastly, the authors thoughtfully discuss emerging challenges including tumour heterogeneity, therapy resistance, biomarker sensitivity, and trial design flaws (e.g., lack of stratification).

Comments

Major:

1. Compared to the robust discussion of AR pathway and cell surface targeting, the metabolic section (e.g., lipid and amino acid metabolism) is underdeveloped. I would like to see the inclusion of more detail on clinical trials, molecular targets, and mechanistic pathways to improve balance and completeness.
2. While the manuscript emphasises biomarker-driven therapy, it under-discusses practical limitations of implementing stratification tools such as ctDNA in clinical settings, esp. regarding sensitivity at low tumour burden, panel selection, and longitudinal monitoring feasibility.
3. The reliance on preclinical evidence in parts of the manuscript (notably epigenetic therapies and kinase inhibitors) reduces clinical translatability. Where possible, more emphasis should be placed on early-phase clinical trial outcomes, or limitations should be clearly stated.
4. I think that the repeated use of "Future Perspective" throughout creates structural redundancy. The authors are encouraged to consolidate these insights into a single, integrative "Future Directions" section at the end to improve flow and cohesion.

Minor:

1. There are minor issues with repetition and formatting, including the duplicate trial IDs (e.g., NCT06551324 appears twice under EZH2 inhibitors) in the table provided.
2. Figure legends may be refined to ensure clarity without redundancy.

******* Reviewer's comments *********Referee #1 (Remarks for Author):**

The review by Nouruzi and colleagues is comprehensive, timely, and powerful. It covers a large body of work in a compelling narrative, and not only includes a full list of supporting publications but also complements this with a full list of clinical trials and therefore the review is exhaustive. My comments are largely clarifications and some organizational queries.

We thank Reviewer 1 for their thoughtful and supportive comments. We appreciated the opportunity to address their suggestions, which we believe have further strengthened and clarified the manuscript.

Comments:

1. Add the rationale for developing AR degraders. Is there an anticipated benefit for degrading the AR compared to antagonizing the receptor? Is that going to be an issue - what was set back in EPI-7386? Wouldn't just get AR independent aggressive lineage? Is there a risk of rapidly developing AR independent aggressive lineages? What was set back in EPI-7386?

Thank you for this thoughtful suggestion. Accordingly, we have expanded our discussion on AR PROTACs to better clarify the rationale behind developing AR degraders and to address the potential challenges associated with this approach. We also clarified what we meant by the "setback" in the case of EPI-7386, referring specifically to the termination of clinical trials due to a lack of improved efficacy over enzalutamide (ENZ). The revised section now reads:

"Other modalities to target AR are under development. For instance, targeting the AR N-terminal domain (NTD) offered an alternative strategy. EPI-7386 showed acceptable safety profiles in phase I trials (NCT04421222, NCT05075577), but phase II trials in combination with ENZ in mCRPC patients were terminated due to a lack of improved efficacy over ENZ alone. This drug is also being evaluated in combination with ENZ in mCSPC patients (NCT06312670) \ Nevertheless, efforts to develop next-generation NTD-targeting agents are ongoing and may yet provide new avenues for overcoming resistance. The next generation of AR-targeted therapies is AR degraders, which should exhibit superior AR pathway inhibition compared to AR antagonists, combat AR-mediated resistance mechanisms, and offer a third line of ARPI following progression on ENZ/ABI. Currently, AR PROTACs are under evaluation in phase I and III in mCRPC (NCT05177042, NCT06764485). However, it is still unclear whether such potent ARPI will manifest in higher rates of AR-independent tumours."

2. Given the AR is a pro-differentiation factor, why is the BAT trial counterintuitive?

Thank you for raising this insightful point. AR is coopted for oncogenic pathways in the context of tumours, which is why all previous therapies have been focused on reducing its activity. BAT is counterintuitive because it tries to push AR back into a pro-differentiation role by overloading it with ligands.

3. Future Challenges on AR therapies - perhaps also mention fragmentomic approaches to develop liquid biopsy that will give better/comprehensive evaluations of AR targeted therapies (i.e. monitor which enhancers are being targeted).

Thank you for the suggestion. Fragmentomic approaches are indeed an important emerging tool, and we have discussed them at the end of the Epigenetic section: "Furthermore, ctDNA fragmentation profiles could be used to quantify the proportion of PRAD and NEPC in patient tumours (De Sarkar,

Patton et al., 2023). However, it is important to note ctDNA alterations may not reflect the true transcriptomic state of the tumour, and other complementary methods should be employed concurrently.”. However, as suggested we added the following to the end of AR therapies section “A key tool that will help to overcome this challenge will be routine use of circulating tumor DNA (ctDNA) sequencing to track the emergence of AR-independent lineage and adjust treatments accordingly (De Sarkar, Patton et al., 2023).”

4. In discussing the identification of actionable genomic alterations it might be useful to highlight the challenges of developing comprehensive catalogs (tissue access, timing etc).

Thank you for the suggestion. As recommended, we have added the following sentence to the end of the first paragraph in the Actionable Genomic Alterations section to acknowledge key challenges: “Identification of these targets is limited by insufficient sampling, tissue access, and timing with respect to treatment.”

5. On the section of PARPi sensitivity there seems to be a disconnect between tumors that are sensitive and the levels of key mutations? Is this the case? Does the concept of "BRCAness" explain how much sensitivity occurs independent of altered BRCA1/2 mutation status for example? Is that a concept?

Thank you for raising this important point. We have addressed it by adding the following sentence to the PARPi sensitivity section: “Furthermore, other alterations may induce a state of BRCAness, which could lead to PARPi sensitivity (Xavier, Rezende et al., 2021), potentially explaining why some HRR-proficient patients still derive benefit. A more comprehensive suite of biomarkers will be needed to expand the pool of patients likely to respond to PARPi.”

6. Why is it surprising that TRIUMPH was null? Clarify why PARPi without ADT had no benefit - is that added? Or is there pathway convergences? Is there a rationale explanation? Likewise can the authors speculate on what the functional assay would be used to stratify HRR activity more precisely.

There are likely multiple factors contributing to the null result in TRIUMPH. First, AR pathway inhibition (ARPI) appears to induce a state of BRCAness, potentially enhancing PARPi sensitivity; without ARPI, this effect may be absent. Second, prostate cancer growth remains heavily dependent on androgen signaling, so in the absence of ADT, tumors may continue to proliferate despite accumulating DNA damage from PARPi treatment.

As for stratifying HRR activity, we mention the potential role of ctDNA-based approaches in the text. However, beyond BRCA1/2 alterations, we are currently limited by the lack of validated functional assays or biomarkers for precise HRR stratification.

We have added the following statement in the second paragraph of HRR section: “In vitro experiments have shown that AR inhibition induces an HRRd like phenotype, offering a potential mechanistic link to why concurrent ARPI is required for PARPi efficacy (Agarwal, Zhang et al., 2023). “

7. Are there molecular profiles that capture high MSI patients? Are they consistent with other tumors where ICB is effective? Are there lessons to be drawn?

Yes, MSI-high patients exhibit a distinct genomic signature that can be identified through tissue or ctDNA sequencing. This MSI signature is a pan-cancer phenomenon and is consistent across tumour types where ICB has shown efficacy, which is why ICB was approved for prostate cancer even in the absence of a dedicated trial specifically in MSI-high PCa patients.

8. Add a line to explain what CDK12 is and why it links to HR deficiency.

As recommended, we have added the following sentence at the beginning of the CDK12 section to clarify its role and link to homologous recombination deficiency: “CDK12 is a cyclin-dependent kinase involved in transcriptional regulation of DNA damage response genes, particularly those required for homologous recombination repair (HRR), such as BRCA1 (Blazek, Kohoutek et al., 2011; Wu, Yu et al., 2023).”

9. More generally, what are the top muts/alterations in advanced PCa, and how many are currently being targeted? (add to F1? Or table?).

Thank you for this valuable suggestion. As recommended, we have expanded our discussion with the following addition: “For instance, the top alterations in advanced prostate cancer include AR, TMPRSS2-ERG fusions, TP53, PTEN, RB1, and FOXA1. However, our understanding of the dependencies associated with these genomic alterations remains limited, and only a small fraction are currently actionable.” Given that not all of these alterations are targetable, we focused this review on those with known inhibitors or associated vulnerabilities. The figures and tables were designed to reflect this specific focus.

10. The section of immune-based therapies is interesting, but what comes through is the lack of success in this arena. In the future perspective perhaps the authors could also add some sense of at what point does it appear that the scientific community have undertaken sufficient work to accept that they cannot reject the null hypothesis, ie that PCa is not sensitive to immunotherapy? Or what would that debate look like? What would be the burden of endpoints?

Thank you for this thoughtful and challenging question. We agree that the limited success of immune-based therapies in prostate cancer raises important questions. However, we believe it is premature to accept the null hypothesis that prostate cancer is inherently insensitive to immunotherapy. At present, we cannot adequately explain why a prostate cancer cell with a certain tumour mutational burden (TMB) is unresponsive to immune checkpoint blockade, while a renal cancer cell with similar features responds (renal cancer has very low TMB, yet immunotherapy is its standard of care). This suggests that the failure of immunotherapy in PCa may reflect an incomplete understanding of the tumour microenvironment and immune context, rather than intrinsic resistance. In this regard, we believe further mechanistic and translational research is still needed before such a conclusion can be drawn.

11. What is rationale for order of sections (# of trials? # approved drugs/strategies? Etc?). An overall figure showing all the different strategies would help the reader (like the last figure, but as a cell-based cartoon?) The rationale for the order of sections in the review was to reflect the clinical and research landscape in a way that guides the reader from current standards to emerging areas. We began with the mainstream and standard-of-care treatments, namely AR-targeted therapies, followed by other clinically implemented strategies such as radiotherapy and PARP inhibitors. We then covered areas with substantial research and clinical trial activity, such as epigenetic and immune-based therapies, even though these have seen more failures than successes. Finally, we highlighted underexplored but promising areas like metabolomics, which we believe hold great potential despite limited research and clinical trial representation.

We appreciate the suggestion to include an overall summary figure. We agree that a schematic, cell-based cartoon showing all therapeutic strategies could help orient the reader and added this in a revised version as Figure 1.

Figure 1. Therapeutic strategies targeting prostate cancer (PCa) vulnerabilities. A schematic representation of diverse treatment approaches in PCa. Current and upcoming key therapeutic strategies include androgen receptor (AR)-targeted therapy, radiotherapy and radiation, precision therapy exploiting DNA repair deficiencies and other actionable mutations, immune checkpoint blockade and adoptive T cell therapies, targeting chromatin modifiers and epigenetic regulators, antibody–drug conjugates, BiTEs, and radioligands directed against cell surface proteins, blocking oncogenic signaling cascades involving kinases, and disruption of metabolic pathways.

12. For the cell surface-based approaches (e.g. ADC) is there any sense for when a marker is targeted successfully (perhaps PSMA) and when not - (perhaps STEAP1) - more generally, is it type of marker - how much is external and how cycled? Docking studies? Any rationale clues?

There are indeed multiple factors that influence the success of an ADC, including the choice of target antigen, its cell surface abundance and internalization rate, as well as antibody specificity and avidity, linker stability, and payload potency. These detailed parameters are beyond the scope of this review, so we focused on summarizing the clinical successes and failures of currently available or investigational ADCs. Other groups, such as Dr. Beltran have recently published a more in-depth discussion of these mechanistic factors (PMID 40169837).

13. Combinatorial ADC/RLTs - how decide which combination? Or is it investigator-based rationale? Currently, combinatorial strategies are largely based on investigator rationale and preclinical data. However, we envision a future approach where ctDNA sequencing is used to define the molecular phenotype of a patient's tumor(s), enabling a tailored therapeutic cocktail that targets multiple vulnerabilities specific to that profile.

14. P.12 - MAPK pathway - the (REF) needs to be replaced and (329133968) in CDK section

We apologize for the error, and the placeholder "(REF)" and reference ID "(329133968)" have now been replaced with the appropriate citations.

15. The epigenetic section (perhaps epigenomic?) after AR may make more sense?

Thank you for the suggestion. We have chosen to retain the current order of sections, as it reflects the progression from standard-of-care therapies to areas with increasing research focus and emerging potential. We hope this structure provides a clear and logical flow for the reader.

16. In the epigenetic section, perhaps clarify in what ways histone states and DNA methyl differ between PRAD and NEPC (where, gain, loss, etc) and illustrate how that contributes to differences?

Thank you for the suggestion. While there are significant differences in histone states and DNA methylation between PRAD and NEPC, the precise regional patterns are beyond the scope of this review. Our focus is on highlighting the existence of these epigenetic differences, their potential utility for patient stratification using emerging technologies, and the clinical relevance of epigenetic inhibitors currently under investigation.

Referee #2 (Remarks for Author):

Nouruzi et al. discuss the main treatment options for prostate cancer (PC) across its different stages, ranging from localized disease to neuroendocrine prostate cancer (NEPC) and double-negative prostate cancer (DNPC). The manuscript primarily focuses on therapies evaluated in clinical trials, while also including some that have shown promising results in preclinical studies. Throughout the article, the authors emphasize the importance of personalized treatment approaches considering the disease's heterogeneity. Notably, the manuscript goes beyond a simple summary of the literature and provides expert insights into the key questions and challenges for the future of the field.

We thank Reviewer 2 for their thoughtful summary and positive assessment of our manuscript. We greatly appreciate their recognition of our effort to provide not only a comprehensive overview of prostate cancer therapies but also the perspective on the challenges and future directions in the field.

I have a few suggestions to further strengthen the manuscript:

1. Define androgen deprivation therapy (ADT), as this will benefit readers who are not specialists in prostate cancer.

Thank you for the suggestion. As recommended, we have added the following definition at the beginning of the Advances in Targeting the AR Pathway section: "Androgen deprivation therapy (ADT) reduces circulating androgen levels or blocks AR activity to inhibit the growth of androgen-dependent prostate cancer. This can be achieved through surgical castration (bilateral orchiectomy), medical castration using gonadotropin-releasing hormone (GnRH) analogs or antagonists, or the use of AR pathway inhibitors. ADT is typically the first-line systemic treatment for prostate cancer."

2. List all authors in the bibliography to help readers identify the research groups and laboratories. We have updated the bibliography to list all authors for each reference, in accordance with EMBO guidelines.

3. Because there is an extensive literature on preclinical studies in PC, it would be helpful if the authors specify in the review the criteria that they used for selecting the preclinical studies that were included.

Thank you for this thoughtful comment. Our selection criteria primarily focused on clinical studies. Preclinical studies were included selectively, not comprehensively, and were used to illustrate proof-of-

concept for particular therapeutic strategies. We did not deliberately exclude any preclinical studies, but rather highlighted those that we felt presented the strongest data or were most translationally relevant or proximal to clinical application.

4. Consider including a section on stereotactic body radiation therapy (SBRT) and cite recent relevant literature, such as: Nikitas J. JAMA Oncol. 2025 May 15:e251059. doi: 10.1001/jamaoncol.2025.1059.

Thank you for the suggestion. As recommended, we have added the following to the beginning of the Radiotherapy and Radiation Therapy section: "Radiotherapy is a mainstay treatment modality for localized prostate cancer (PCa). For low-risk patients, brachytherapy is a common choice (Peinemann, Grouven et al., 2011), while hypofractionated radiotherapy and stereotactic body radiotherapy (SBRT) offer shorter, yet effective, treatment durations. These innovations are expanding the range of treatment options, making radiotherapy a versatile approach for managing PCa (van As, Griffin et al., 2024; Widmark, Gunnlaugsson et al., 2019). A recent clinical trial demonstrated that SBRT was well tolerated after radical prostatectomy compared to conventionally fractionated radiotherapy (Nikitas, Ballas et al., 2025). Radiotherapy is also used in certain advanced PCa cases."

Referee #3 (Remarks for Author):

This is a timely, well-structured, and deeply researched review which appears to be within the scientific scope of your journal. The manuscript provides valuable insights into prostate cancer (PCa) therapy, from its molecular mechanisms to clinical applications.

The article provides a thorough and structured overview of the evolving PCa treatment landscape, spanning conventional systemic therapies (ARPI, chemotherapy, radiotherapy) to emerging targeted strategies (ADCs, RLTs, immunotherapy, metabolic and epigenetic approaches). It also nicely captures the complexity of AR-driven and AR-independent subtypes (e.g., NEPC, DNPC), and articulates resistance mechanisms and lineage plasticity.

The review effectively links biomarkers to clinical applications, thus underscoring the translational significance of genetic stratification and ctDNA analysis. The discussion of ctDNA-based monitoring is appropriately critical. The inclusion of ongoing clinical trials (with trial IDs and disease settings) enhances its translational depth. Tables and figures summarising trial data and therapeutic mechanisms are helpful for both researchers and clinicians working in this field. Importantly, the review covers not only what is approved, but also what is investigational, clearly positioning the trajectory of this field. Lastly, the authors thoughtfully discuss emerging challenges including tumour heterogeneity, therapy resistance, biomarker sensitivity, and trial design flaws (e.g., lack of stratification).

We sincerely thank Reviewer 3 for their thoughtful and encouraging comments. We are grateful for their recognition of the manuscript's depth, structure, and translational relevance, as well as their appreciation of our efforts to integrate molecular insights with clinical applications. Their feedback reinforces our goal of providing a comprehensive and forward-looking overview of the evolving prostate cancer treatment landscape.

Comments

Major:

1. Compared to the robust discussion of AR pathway and cell surface targeting, the metabolic section (e.g., lipid and amino acid metabolism) is underdeveloped. I would like to see the inclusion of

more detail on clinical trials, molecular targets, and mechanistic pathways to improve balance and completeness.

We agree that the metabolic section is less developed compared to others, which reflects the early and still evolving nature of this field in cancer therapeutics, including in prostate cancer. The role of the metabolome remains underappreciated, and while there are several clinical trials involving dietary interventions, many focus on quality-of-life outcomes rather than enhancing treatment response or targeting defined metabolic pathways. Additionally, most did not incorporate metabolic endpoints or were too general (e.g., weight loss diets) to draw mechanistic conclusions. Therefore, we chose to highlight trials that directly target specific metabolic vulnerabilities in PCa with cancer-specific outcomes, though such trials remain limited at this time.

2. While the manuscript emphasises biomarker-driven therapy, it under-discusses practical limitations of implementing stratification tools such as ctDNA in clinical settings, esp. regarding sensitivity at low tumour burden, panel selection, and longitudinal monitoring feasibility.

We agree that the practical limitations of implementing ctDNA-based stratification in clinical settings are critical to consider. These challenges, were discussed in the final paragraph of the Genomic Alterations section: “While ctDNA offers a powerful, non-invasive approach to detect and characterize patient tumours, there are still limitations. Some patients with low or undetectable ctDNA can still have cancer cells present that will progress to mCRPC. It is challenging to determine alterations in the tumour genome with low ctDNA fractions (< 2%) [81]. Alteration detection is further limited by the sequencing methodology used, such as targeted panels, whole-exome and whole-genome sequencing. ctDNA may also miss some alterations that would otherwise be detected by sequencing the tumour tissue directly [82, 83]. Ultimately, multi-modal approaches will be required to characterize tumours and guide optimal treatment options. ctDNA should be complemented with transcriptomic, IHC, or radiographical imaging, including PET [84, 85]”. We also briefly reiterated these limitations in the Epigenomics section to emphasize the need for integrative strategies in clinical decision-making: “However, it is important to note ctDNA alterations may not reflect the true transcriptomic state of the tumour, and other complementary methods should be employed concurrently.”

3. The reliance on preclinical evidence in parts of the manuscript (notably epigenetic therapies and kinase inhibitors) reduces clinical translatability. Where possible, more emphasis should be placed on early-phase clinical trial outcomes, or limitations should be clearly stated.

Thank you for this thoughtful comment. To the best of our ability, we aimed to highlight the successes and failures of clinical trials across all therapeutic areas. In sections such as epigenomics, kinase inhibitors, and metabolomics, the number of relevant clinical trials, whether successful or not, remains limited. As such, we included preclinical evidence selectively to illustrate proof-of-concept for emerging treatment strategies. Where clinical data were lacking, we aimed to clearly reflect this and to frame the preclinical studies as foundational rather than definitive.

4. I think that the repeated use of "Future Perspective" throughout creates structural redundancy. The authors are encouraged to consolidate these insights into a single, integrative "Future Directions" section at the end to improve flow and cohesion.

Thank you for the helpful suggestion. We agree that repeated "Future Perspective" subheadings could create structural redundancy. These sections were originally intended to serve as brief conclusions for each therapeutic section, given the breadth of the review. However, based on your feedback, we have reformatted the manuscript to integrate these insights as concluding paragraphs within each section, rather than as separate subheadings, to improve overall flow and cohesion.

Minor:

1. There are minor issues with repetition and formatting, including the duplicate trial IDs (e.g., NCT06551324 appears twice under EZH2 inhibitors) in the table provided.

We apologize for the oversight and thank you for bringing it to our attention. The duplicate trial IDs and formatting issues have been corrected accordingly.

2. Figure legends may be refined to ensure clarity without redundancy.

We have revised the figure legends to enhance clarity and remove redundancy, as requested.

16th Jun 2025

Dear Amina,

Thank you for sending your revised manuscript. I am pleased to inform you that your manuscript is now accepted for publication and will be sent to our publisher as soon as the figures will be redrawn and approved by you. Our graphic designer will be in touch when the figures are ready.

Your manuscript will be processed for publication by EMBO Press. It will be copy edited and you will receive page proofs prior to publication. Please note that you will be contacted by Springer Nature Author Services to complete licensing information.

There is no charge for this Review Article, but in a few weeks when you are contacted to sign your license agreement and review article proofs, please enter this token into the appropriate field in the Springer Nature author services system: *Token unavailable*.

If you have any questions, please do not hesitate to contact the Editorial Office.

Thank you for your contribution to EMBO Molecular Medicine!

With kind regards,

Lise
